# Inhibition of cytosine 5-hydroxymethylation during progression of cancer precursor lesions in the uterine cervix

**Jobran M. Moshi[1,2], Monique Ummelen[1], Frank Smedts[3], Frans C. S. Ramaekers[1], Anton H. N. Hopman[1]** *

1 Department of Molecular Cell Biology, GROW-School for Oncology and Reproduction, Maastricht University Medical Center, Maastricht, the Netherlands, 2 Department of Medical Laboratory Technology, Faculty of Applied Medical Sciences, Jazan University, Jazan, Kingdom of Saudi Arabia, 3 Department of Pathology, Cork University Hospital, Cork, Ireland

* hopman@maastrichtuniversity.nl

**Data Availability Statement:** All relevant data are within the paper and its Supporting Information files.

## Abstract

Methylation and hydroxymethylation of cytosine moieties in CpG islands of specific genes are epigenetic processes shown to be involved in the development of cervical (pre)neoplastic lesions. We studied global (hydroxy)methylation during the subsequent steps in the carcinogenic process of the uterine cervix by using immunohistochemical protocols for the detection of 5-methylcytosine (5-mC) and 5-hydroxymethylcytosine (5-hmC) in paraffin-embedded tissues of the normal epithelia and (pre)malignant lesions. This approach allowed obtaining spatially resolved information of (epi)genetic alterations for individual cell populations in morphologically heterogeneous tissue samples. The normal ectocervical squamous epithelium showed a high degree of heterogeneity for both modifications, with a major positivity for 5-mC in the basal and parabasal layers in the ectocervical region, while 5-hmC immunostaining was even more restricted to the cells in the basal layer. Immature squamous metaplasia, characterized by expression of SOX17, surprisingly showed a decrease of 5-hmC in the basal compartments and an increase in the more superficial layers of the epithelium. The normal endocervical glandular epithelium showed a strong immunostaining reactivity for both modifications. At the squamocolumnar junctions, a specific 5-hmC pattern was observed in the squamous epithelium, resembling that of metaplasia, with the typical weak to negative reaction for 5-hmC in the basal cell compartment. The reserve cells underlying the glandular epithelium were also largely negative for 5-hmC but showed immunostaining for 5-mC. While the overall methylation status remained relatively constant, about 20% of the high-grade squamous lesions showed a very low immunostaining reactivity for 5-hmC. The (pre)malignant glandular lesions, including adenocarcinoma in situ (AIS) and adenocarcinoma showed a progressive decrease of hydroxymethylation with advancement of the lesion, resulting in cases with regions that were negative for 5-hmC immunostaining. These data indicate that inhibition of demethylation, which normally follows cytosine hydroxymethylation, is an important epigenetic switch in the development of cervical cancer.

**Funding:** Jobran Moshi: His work was supported by a grant from Jazan University, Jazan, Saudi Arabia (grant number 1067568103). This study was part of the PhD study of Jobran Moshi. The funders had no role in study design, data collection and analysis, decision to publish, or preparation of the manuscript.

**Competing interests:** The authors have declared that no competing interests exist.

## Introduction

Genomic methylation is involved in the regulation of a plethora of epigenetic processes in both prokaryotes and eukaryotes. In prokaryotes, adenine is the most important methylation target, while such modifications in mammalian cells mostly involve cytosine methylation of CpG residues to create 5-methylcytosine (5-mC). CpG dinucleotides are found across the entire genome and are often clustered in islands where the cytosine is methylated. CpG DNA methylation significantly changes during normal growth and differentiation, but also during ageing and cancer formation. Active demethylation has been shown to be mediated by oxidation of 5-mC by ten-eleven translocation (TET) proteins, which convert 5-mC into 5-hydroxy-methylcytosine (5-hmC) [1, 2].

The presence of 5-mC has been described in several normal tissues such as muscle, lung, kidney, heart, brain, gastric tract and embryonic cells [3–7]. Numerous genes have been found to undergo hypermethylation in cancer. The genes that are susceptible are the genes involved in cell cycle regulation (e.g. *p16INK4a*, *Rb*), genes associated with DNA repair (e.g. *BRCA1*, apoptosis (*DAPK*, *TMS1*), drug resistance, detoxification, differentiation, angiogenesis, and metastasis. Although certain genes such as *RASSF1A* and *p16* are commonly methylated in a variety of cancers, other genes are methylated in specific cancers [8, 9].

Usually, during carcinogenesis, the methylation level around specific genes is increased. Still, global methylation of the genome is low because cells have a natural tendency to normalize methylation via the active demethylation mechanism described above [10]. Several studies have examined 5-mC as a global DNA methylation marker in different cancer types such as cervix, colon and oral cavity malignancies [11, 12]. In addition, 5-hmC has been detected in different tissues [13] and cancer types such as those from liver, lung, pancreas, breast, brain, prostate and ovary [4, 5, 14].

To date, several publications have indicated that 5-mC and 5-hmC may become diagnostic and prognostic biomarkers for cervical cancer [15–17]. In addition, several studies have reported that the methylation and demethylation processes can impact the origin and progression of pre-malignant lesions, especially those in the cervix [11, 12]. Cervical cancer, with human papillomavirus (HPV) infections being the primary causal factor, is the second most frequent gynecological cancer among women globally. In 2018 the prevalence of cervical cancer rate was 570,000 new cases, which increased to reach 604,000 new cases in 2020 despite the introduction of the HPV vaccination program in several countries worldwide [18, 19]. One model for the carcinogenic process involves HPV infection of metaplastic squamous epithelium or the cervical reserve cells, which are believed to be a progenitor cell population capable of generating squamous-type epithelium. Besides that, these cells can differentiate into columnar cells, thereby replenishing the endocervical glandular epithelium. Upon persistent HPV infection, these different cell and tissue types can produce cervical intraepithelial neoplasms (CIN lesions) and finally invasive squamous cell carcinoma or adenocarcinoma in situ (AIS) and finally adenocarcinoma [20, 21].

The stepwise process of carcinogenesis can be ideally studied in the uterine cervix, where subsequent stages of metaplastic, dysplastic and (micro)invasive changes can be easily accessed by biopsies of the transformation zone (TZ) and then be studied in a histo-morphological context. Since several studies have shown that multiple genes are hypermethylated during the process of cervical carcinogenesis (for an overview see [4, 22–24]), we wondered whether the detection of the global methylation and hydroxymethylation status could be used as indicators of epigenetic switches occurring during cervical carcinogenesis. Using an immunohistochemical approach we obtained spatially resolved information that allowed us to correlate the (hydroxy)methylation status to the position of individual cells in morphologically heterogeneous tissue samples derived from normal cervix and its (pre)malignant derivatives.

## Materials and methods

### Tissues

In this retrospective study of archived samples, all data were fully anonymized before the study was started. The ethics committee waived the requirement for informed consent. Formalin-fixed and paraffin-embedded (FFPE) tissue samples of the uterine cervix, obtained after colposcopy of 60 patients with cytological indications of a (pre)malignant lesion (Table 1), were selected from the archives of the Departments of Pathology of the SSZOG (Stichting Samenwerking Ziekenhuizen Oost Groningen) and of the Reinier de Graaf Hospital Delft. These tissues are used in the context of routine diagnostics, after which they are archived. Samples were collected between 2008 and 2020. Research on these anonymized archival tissue samples was performed in accordance with the Declaration of Helsinki and with the Code for Proper Secondary Use of Human Tissue in the Netherlands (http://www.federa.org/, update 2011: Human Tissue and Medical Research).

### Tissue processing for immunohistochemistry

Four μm thick paraffin sections were collected on Superfrost Plus microscope slides (Thermo Fisher Scientific, Waltham, Massachusetts, United States), dewaxed in xylol, rehydrated in a descending alcohol series and processed further using the citrate retrieval protocol. For comparison of staining intensities, a few slides of squamous and glandular epithelia were subjected to the Tris/EDTA pH 9.0 (TE) or Pepsin/HCl antigen retrieval protocol as described before [25]. For the citrate protocol, the specimens were boiled for 20 minutes in 10 mM Na-citrate buffer (pH 6.0) in a microwave oven and then left at room temperature for 30 minutes to cool down. Subsequently, slides were 3 times dip washed with demineralized water. Before starting with the immunohistochemical incubations the specimens were washed with PBS/0.05% Tween-20 (PBST) for 5 min.

### Antibodies and immunohistochemical protocols

The following primary antibodies to 5-mC and 5-hmC were applied in this study:

**Table 1. Selected formalin-fixed and paraffin embedded tissue samples of (pre)malignant lesions of the uterine cervix.**

|  | Patient diagnosis* | Nr of patients | Analyzed histological areas | Areas analyzed |
|---|---|---|---|---|
|  |  |  | Normal squamous epithelium[#] | 73 |
|  |  |  | Normal glandular epithelium[#] | 26 |
|  |  |  | Normal SCJ | 8 |
| Squamous lesions | CIN 1 | 6 | CIN 1 | 8 |
|  | CIN 2 | 3 | CIN 2 | 7 |
|  |  |  | CIN 1[+] | 1 |
|  | CIN 3 | 25 | CIN 3 | 55 |
|  |  |  | <CIN 3[+] | 22 |
| Glandular lesions | AIS | 12 |  | 21 |
|  | AdC | 14 |  | 18 |
|  | Total | 60 |  | 239 |

*: Patients were diagnosed on basis of the most severe CIN grade present in the lesion.

#: No healthy controls were examined, normal histological areas present in the biopsies were analyzed. These areas were present in the patient samples and recognized by the pathologist and by molecular analysis. The normal areas showed to be HPV and p16 negative.

+: coinciding areas with less severity. CIN: cervical intraepithelial lesion; AIS: adenocarcinoma in situ; AdC: adenocarcinoma.

SCJ: normal squamocolumnar junction (transition between squamous epithelium and glandular/columnar epithelium), no preneoplasia present in the junction.

1. The mouse monoclonal OptimAb anti 5-methylcytosine (anti 5-mC), (BI-MECY-0100, clone 33D3; Eurogentec, Seraing, Belgium). Clone 33D3 recognizes the modified base 5-methylcytidine in DNA and specifically distinguishes it from its normal DNA base counterpart.

2. The recombinant rabbit monoclonal anti-5-hydroxymethylcytosine antibody (anti 5-hmC) (clone RM236, ab214728; Abcam, Cambridge, UK). The antibody ab214728 reacts to 5-hydroxymethylcytosine in both single-stranded and double-stranded DNA. No cross reactivity with non-methylated cytosine and methylcytosine in DNA is reported.

These two antibodies were diluted 1:100 and detected in single- and double-label immuno-fluorescence protocols as detailed in S1 Table (S1A), as well as in a brightfield immunohisto-chemical protocol as detailed in S1 Table (S1B).

As negative controls, incubations were performed with only the secondary antibodies.

Additional antibodies directed against SOX2, SOX17, p16, cytokeratin 7, cytokeratin 17 and Ki-67, as well as optimized detection protocols used for the bright field immunohisto-chemical characterization of the different cell and tissue types in (pre)malignancies are summarized in S1 Table.

## Fluorescence and bright field microscopy

Fluorescence microscopy: After subsequent incubations with the primary and fluorochrome-conjugated secondary antibodies as described before [25], the tissue sections were washed 2 times 5 minutes with PBST and once with PBS for 5 minutes, then dehydrated and mounted with 90% glycerol in 0.02M Tris/HCl (pH 8.0), containing 0.02% NaN3, 2% DABCO and 0.5 µg/ml 4', 6-diamidino-2-phenylindole (DAPI; Merck 10236276001).

For non-confocal imaging the fluorescent signals were recorded with the Metasystems Image Pro System (black and white CCD camera; Sandhausen, Germany) mounted on top of a Leica DM-RE fluorescence microscope, equipped with single band pass (BP) filters for single color analysis of DAPI (excitation BP340-380; emission BP450-490), FITC (excitation; BP460-500; emission BP512-490), Texas Red (excitation BP540-580; emission BP593-668) and TRITC, used to view the Alexa 555 conjugate (excitation BP510-560; emission BP572-647). Images were obtained using an automatic integration time, allowing quantitative evaluation, and using the full dynamic range of the camera without signal intensity saturation [25].

For confocal imaging the sections were imaged with a laser scanning confocal microscope (Leica SPE confocal) using LAS AF software (Leica Application Suite Advanced Fluorescence, version 2.7.3.9723). Imaging was done in acquisition mode xyz, with an ACS APO 63.0x1.30 oil immersion lens, a gain of 800-1000V, offset between -0.5 and 0%, the format of 1024x1024 pixels, speed of 400 Hz and frame average of 3. For DAPI excitation, a laser wavelength of 405 nm was used. FITC was excited at 488nm (emission filter 500–594 nm), while Texas Red and Alexa 555 were excited at 532 nm (emission filter 575–737 nm). Image J (https://imagej.nih. gov/ij/docs/menus/analyze.html; NIH, Bethesda, Maryland, USA) was used for further image analysis, processing and merging of the fluorescent images for the reconstruction of the sections.

Bright field microscopy: After antibody incubations as detailed in S1 Table (S1B) for bright field microscopy, the sections were washed in PBS and incubated with freshly prepared diami-nobenzidine (Liquid DAB+ substrate chromogen system; DAKO, K3467) for 7 minutes at room temperature and subsequently washed with MilliQ water. The sections were counter-stained with hematoxylin (Hematoxylin Solution modified according to Gill II, Merck, 1.05175.0500), dehydrated in an ascending ethanol series and mounted with Entallan (Merck

1.07961.0100). The bright field sections were scanned with a Ventana iScan HT slide scanner (Ventana Medical Systems, Inc. Tucson, Arizona, USA), and images were viewed and selected using Image Viewer Software (Ventana MS).

## Results

For the visualization of global methylation and hydroxymethylation of DNA, the citrate pH 6.0 pretreatment protocol was applied as previously described [25]. In this previous study we have also shown that a wide range of fluorescence intensities was typical for the immunohisto-chemical detection of 5-mC and 5-hmC in normal squamous epithelium. This was observed and quantified by fluorescence intensity measurements in cells belonging to the different compartments of the epithelium, i.e. the basal/parabasal, intermediate and superficial cell layers. Together with the result from simultaneous detection of 5-mC and 5-hmC within individual nuclei, these observations suggested that immunohistochemical detection levels of methylation and hydroxymethylation can not only differ between different cellular compartments but even between neighboring cells within a single cell compartment. In the underlying study we were confronted with the observation that not only differences in signal intensities but also typical differences in staining patterns can be found in both normal and premalignant lesions. Because of this heterogeneous immunofluorescence reactivity noticed for 5-mC and 5-hmC, with cells showing staining for both or only one of the two, 5-mC and 5-hmC were mostly stained simultaneously and are presented in single- and merged-color images for visual inspection. We classified the immunofluorescence intensities for 5-mC and 5-hmC as negative, weak, intermediate or intense/strong, with staining for 5-hmC in stromal tissue always being intense.

### 5-mC and 5-hmC immunostaining levels in normal squamous epithelium

Fig 1A depicts a H&E-stained stretch of normal squamous ectocervical epithelium with underlying stroma. In the right part of the epithelium, a gland is recognized close to the basement membrane (see Fig 1B). In Fig 1C and 1D an overview of the immunofluorescence detection of 5-mC and 5-hmC is shown at low magnifications, while higher resolution immunofluorescence images for the lower compartments of the epithelium are shown in Fig 1E–1K. Immunoreactivity for 5-mC is seen from the basal/parabasal layer up to the superficial layer and small differences in staining intensity were noticed over the length of the stretch of epithelium. Negative and weakly stained nuclei were seen to be intermingled with strongly stained nuclei (Fig 1E, 1H, 1K and 1N).

For 5-hmC, however, differences in intensities were recognized throughout the different layers of the epithelium. Intense staining is frequently seen in the basal cell layer which differs in intensity from the layers above up to the intermediate cell layer (Fig 1C and 1F; this pattern was seen in 21 out of 73 normal areas analyzed). These intensity differences were also found when other pretreatment retrieval procedures were applied to the slides (results not shown). The epithelium in areas more closely located to the squamo-columnar junction (SCJ) showed a more intense 5-hmC staining in the intermediate cell layer as compared to the basal/parabasal layers (Fig 1I and 1L). This weaker immunostaining of 5-hmC in the basal cell compartment of the squamous epithelium, mainly recognized in the areas where glands were underlying the epithelium, instigated us to study in more detail the topological distribution of 5-hmC at the position of the SCJ. Fig 2A and 2B show the H&E-stained junction between squamous and columnar epithelium, with the squamous epithelium adjacent to the SCJ showing intense 5-mC staining from the basal/parabasal compartment up to the superficial layer of the epithelium (Fig 2C and 2G), while for 5-hmC a weak staining is seen in the basal/parabasal

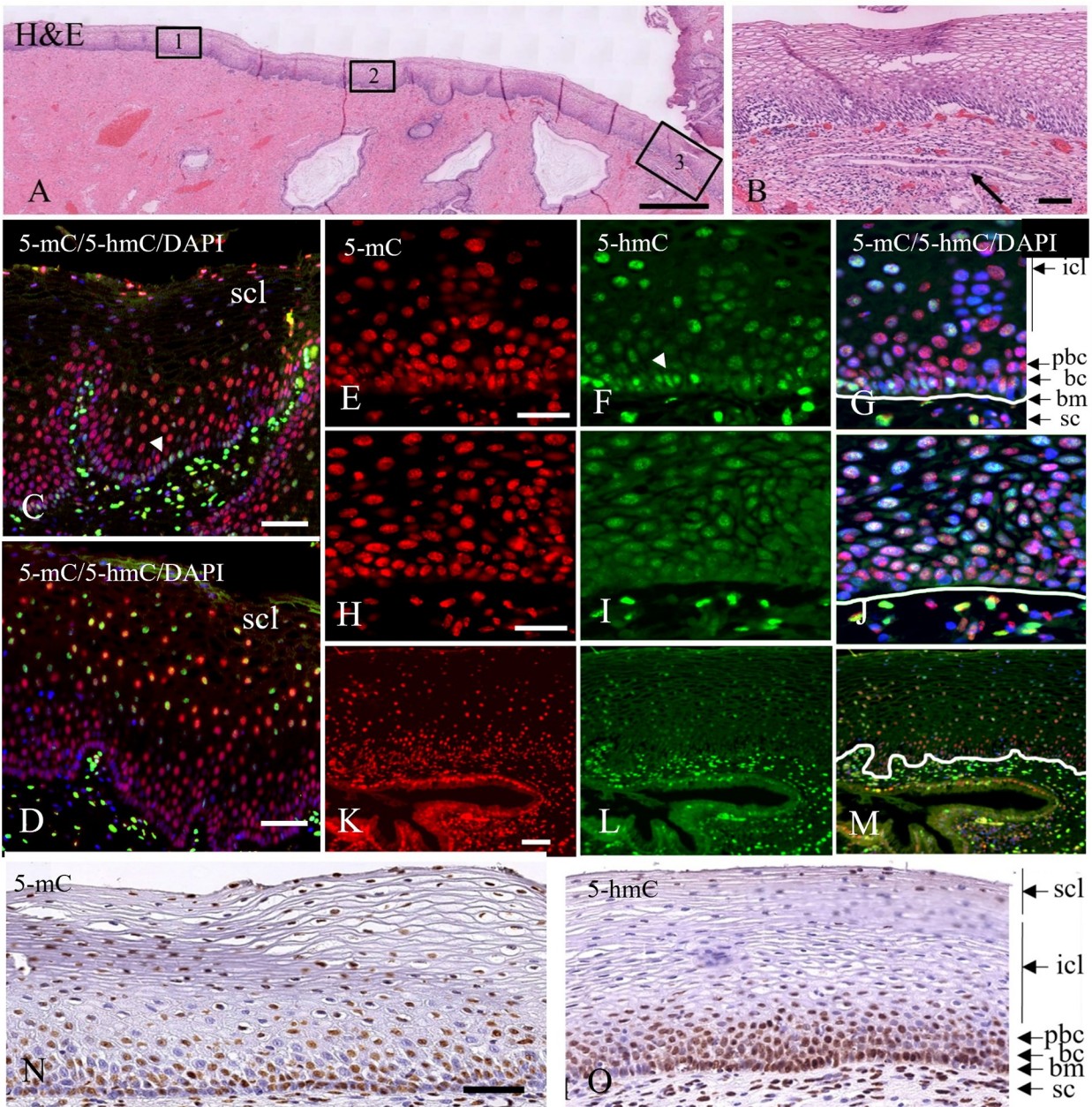

**Fig 1. Double-label immunostaining of 5-mC and 5-hmC in normal squamous epithelium. A)** Low magnification of an H&E-stained stretch of normal squamous epithelium. Boxes 1, 2 and 3 indicate areas referred to below. **B)** Higher magnification of H&E-stained epithelium in box 3 with a gland underlying the normal squamous epithelium (arrow). **C, D)** Low magnification immunofluorescence images of merged staining patterns showing 5-mC in red and 5-hmC in green combined with blue DAPI counterstaining. **E-M)** Higher magnification immunofluorescence images of 5-mC in red (**E, H** and **K**), 5-hmC in green (**F, I** and **L**) and merged images of 5-mC, 5-hmC with blue DAPI counterstaining in **G, J** and **M**. Images **E-G** are taken from box 1, **H-J** from box 2 and **K-M** from box 3, the latter showing the gland also seen in **B**. The lines in **G, J** and **M** mark the position of the basement membrane. **N, O)** Bright field immunohistochemistry of 5-mC (**N**) and 5-hmC (**O**). The stromal cells, basement membrane, basal and parabasal cells, and the intermediate and superficial cell layers are indicated by sc, bm, bc, pb, icl and scl, respectively. Arrow heads in **C** and **F** point to basal cells with a high immunofluorescence staining for 5-hmC. Note also the intense 5-hmC immunostaining in the stromal cell compartment. Scale bars indicate 500 μm in **A**, 100 μm in **B, C** and **D**, 20 μm in **E and H** (same magnifications in **F, G, I**, and **J**), and 100 μm in **K** and **N** (same magnification in **L, M** and **O**).

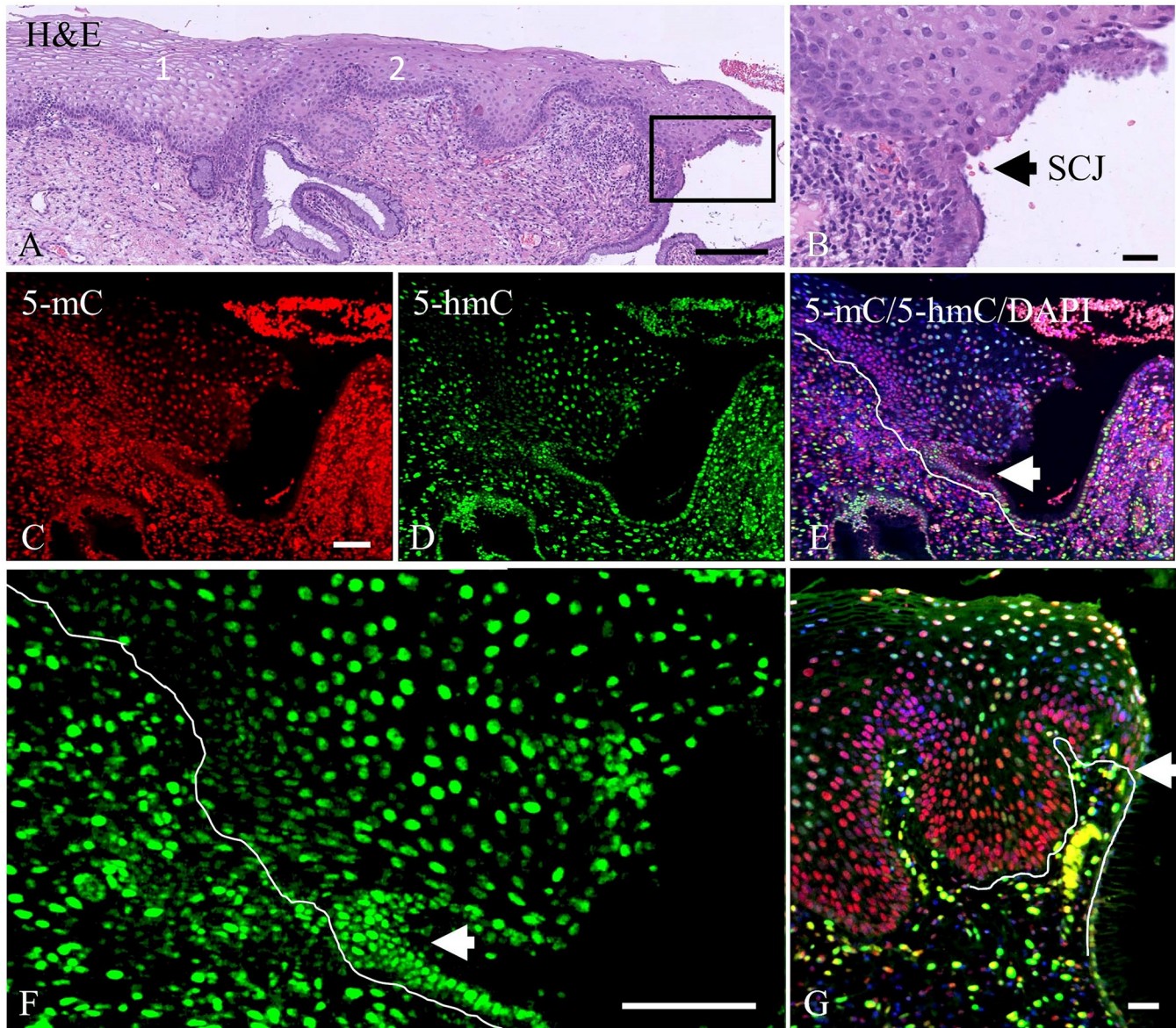

**Fig 2. Double-label immunostaining results for 5-mC and 5-hmC in the squamo-columnar junction (SCJ). A)** H&E-stained stretch of squamous epithelium with SCJ, higher magnification in **B** (arrow). **C-E)** SCJ area shown in box in **A**, with arrows pointing at the junction between squamous and columnar epithelium. In (**C**) red immunofluorescence of 5-mC, (**D**) 5-hmC in green and (**E**) merged images of 5-mC, 5-hmC and blue DAPI counterstaining. **F)** Higher magnification of 5-hmC immunostaining as shown in **D. G)** Merged images for 5-mC (red) and 5-hmC (green) with DAPI counterstaining in another area showing the SCJ. The lines in **E**, **F** and **G** mark the position of the basement membrane. Note also the intense 5-hmC immunostaining in the stromal cell compartment. Scale bars indicate 500 μm in **A**, 100 μm in **C** (same magnifications in **D** and **E**), and 20 μm in **B**, **F** and **G**.

layer compared to the superficial layers (Fig 2D and 2F). Fig 2A also shows that two histologically different types of normal epithelia can be recognized distal from the SCJ (areas 1 and 2), the right part of which represents metaplastic squamous epithelium, which is supported by the presence of a gland underlying the epithelium, indicating a new SCJ. We defined metaplasia as normal epithelium since it represents the transition of endocervical glandular epithelium into squamous epithelium, a normal physiological process that can be detected in cervical biopsies. Metaplastic epithelium characterizes the transformation zone and bridges the epithelium between the original and new SCJ. Fig 3A shows a stretch of (immature) metaplastic

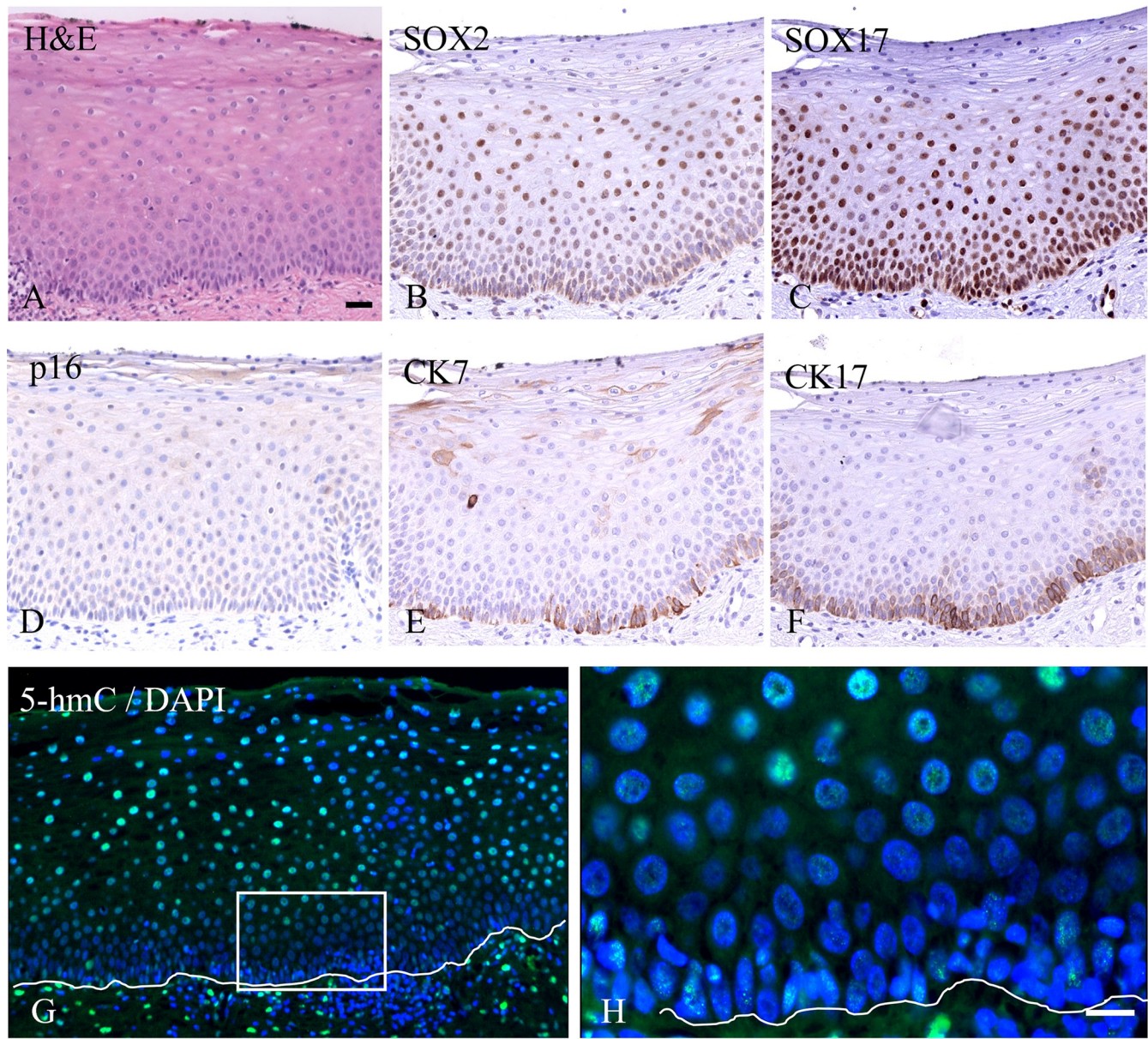

**Fig 3. Immunostaining for 5-hmC in (immature) metaplastic squamous epithelium. A**) H&E staining of metaplastic epithelium with **B**) immunostaining for SOX2, **C**) SOX17, **D**) p16, **E**) cytokeratin 7, and **F**) cytokeratin 17. **G**) Immunofluorescent staining of 5-hmC in green and DNA with DAPI in blue. **H**) Higher magnification of the boxed area indicated in **G**. The lines mark the position of the basement membrane. The scale bar indicates 100 μm in **A** (same magnifications in **B**-**G**) and 20 μm in **H**.

epithelium with the typical co-expression of SOX2 and SOX17 (Fig 3B and 3C, respectively), and at the same time a negative p16 staining (Fig 3D), indicating that the epithelium was not infected by HPV. For an overview showing the transition in these immunostaining patterns between normal and metaplastic epithelium we refer to S1 Fig. Also typical for metaplastic squamous epithelium, cytokeratin7 was expressed in the basal/parabasal and superficial compartments, while cytokeratin17 was found solely in the basal/parabasal layer. Immunostaining for 5-hmC was mainly seen in the intermediate and upper compartment and was weak to

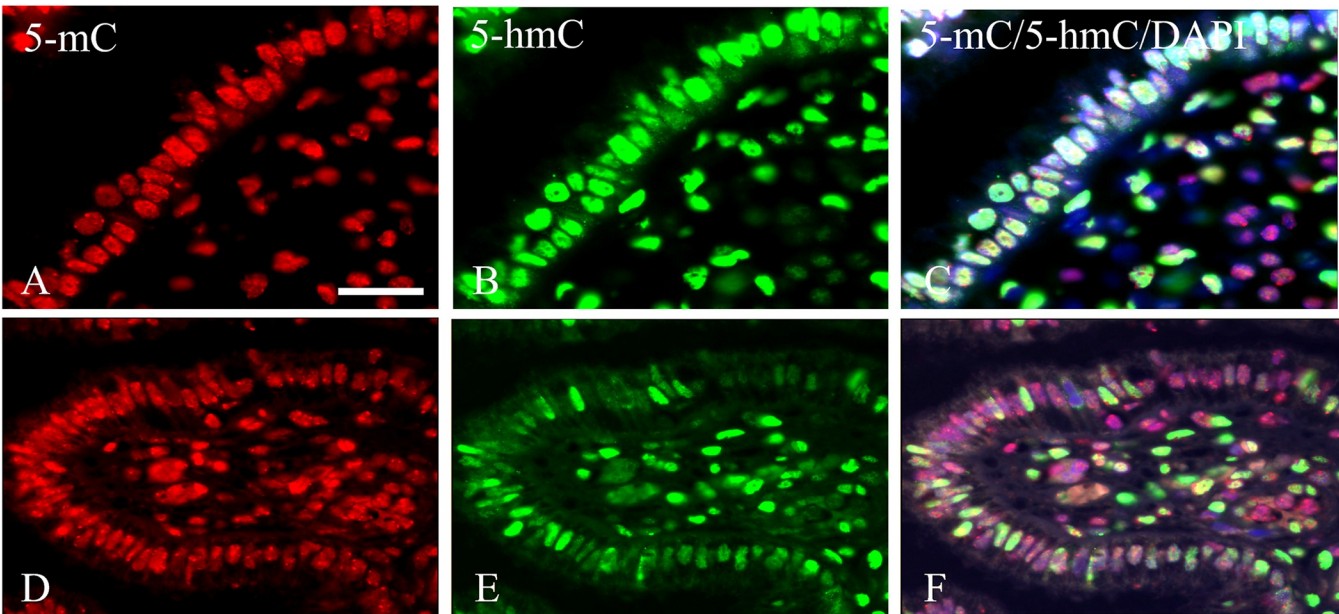

**Fig 4. Immunostaining results for 5-mC and 5-hmC in normal endocervical columnar epithelium. A and D**) 5-mC; **B and E**) 5-hmC; **C and F**) merged images of 5-mC and 5-hmC with DAPI counterstaining. Scale bar in **A** indicates 20 μm (same magnifications in **B**-**F**).

negative in the basal/parabasal layers (Fig 3G and 3H), similar to what was observed in most of the SCJ areas.

## 5-mC and 5-hmC immunostaining levels in the normal columnar epithelium and reserve cells

Intense immunofluorescence staining could be observed for both 5-mC and 5-hmC in normal columnar epithelium (Fig 4). The epithelial nuclei showed a significant heterogeneity in staining along the trajectory of the columnar epithelium, with an even more extensive range in 5-hmC intensities compared to 5-mC. This holds for the columnar epithelium close to and further away from the SCJ.

Reserve cells present in the endocervix beneath the columnar epithelium are considered progenitor cells for the squamous epithelium and can differentiate into (immature) metaplastic squamous epithelium. Fig 5A–5F illustrate the phenotype of these cells with a positive expression profile for SOX17, cytokeratin 7 (CK7) and 17 (CK17) but negative for p16 and SOX2. The columnar epithelial cells are SOX17 and cytokeratin 7 (CK7) positive, but negative for cytokeratin 17 (CK17), p16 and SOX2. Occasionally Ki-67 positive cells could be recognized in the bilayer of columnar and reserve cells. These reserve cells are strongly positive for 5-mC (Fig 5G), but only weak or even negative for 5-hmC (Fig 5H). The merged image of 5-mC, 5-hmC and DAPI clearly shows the methylation and hydroxymethylation level differences in the reserve cells and the overlying columnar cells (Fig 5I).

## Complex immunostaining patterns for 5-mC and 5-hmC in squamous intraepithelial lesions

The patient diagnosis as shown in Table 1 was based on the highest histological CIN grade observed in the tissue sample, but virtually all samples contained areas with different grades of CIN. This was especially obvious in the higher-grade lesions. Using H&E staining of the tissue

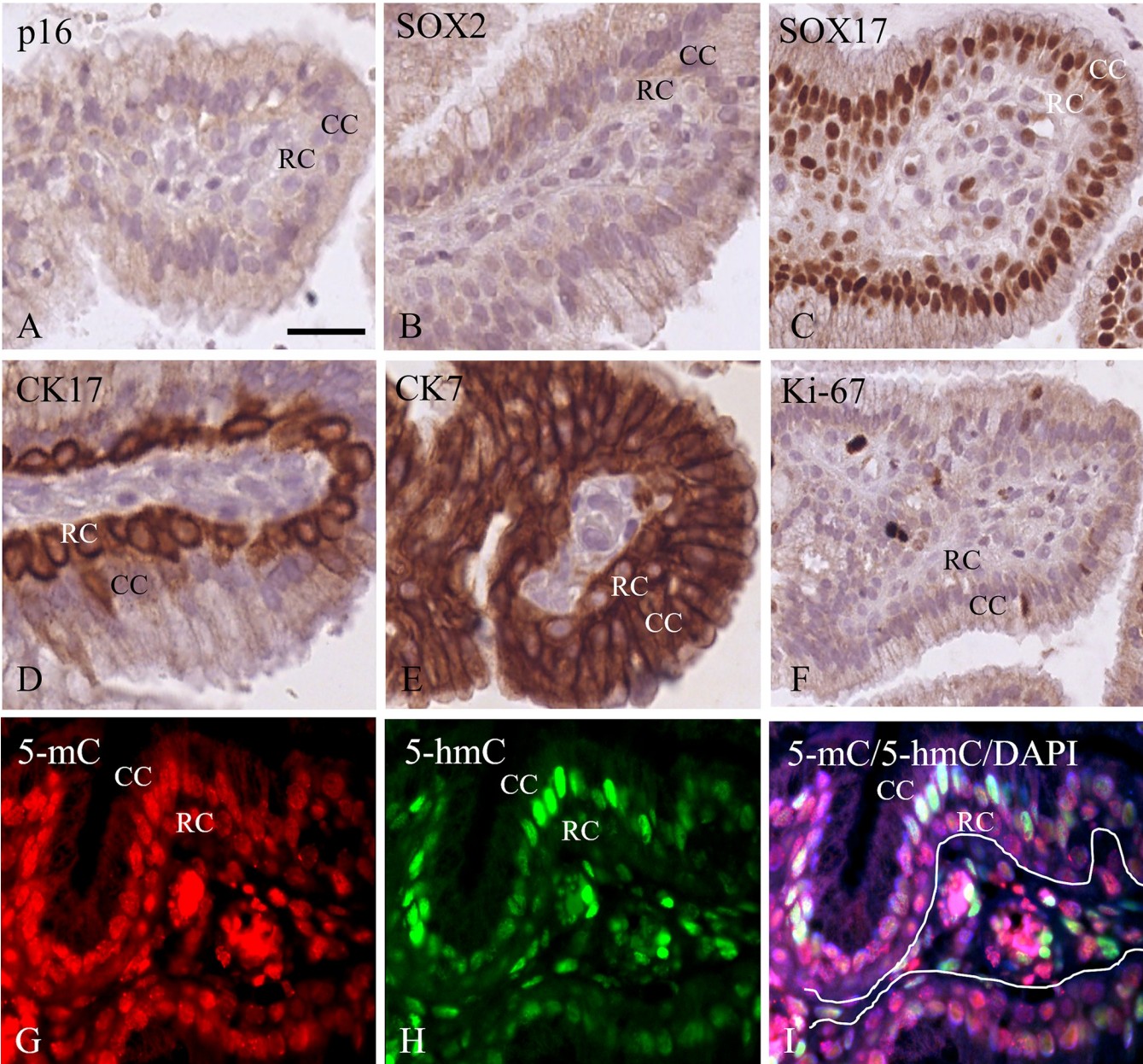

**Fig 5. Phenotypic profile and 5-mC and 5-hmC immunostaining results of normal glandular epithelium and underlying reserve cells. A-F)** Bright field immunostaining of p16 (**A**), SOX2 (**B**), SOX17 (**C**), cytokeratin 17 (CK17) (**D**), cytokeratin 7 (CK7) (**E**) and Ki-67 (**F**). Note the intense staining of the reserve cells for CK17 in **D**, while columnar epithelium and the reserve cells are SOX17 and CK7 positive. **G-I)** Immunofluorescence staining of 5-mC (**G**), 5-hmC (**H**) and merged images of 5-mC and 5-hmC with counterstaining of DNA with DAPI (**I**). The line in **I** marks the position of the basement membrane. Scale bar in A indicates 20 µm (same magnifications in **B-I**). The columnar epithelial cells and reserve cells are indicated as CC and RC, respectively.

sections of the different patient samples as a guide to recognize specific areas with different grades of CIN, a correlation was made between the CIN stages on the one hand and the different immunohistochemical (hydroxy)methylation signal distribution patterns seen throughout the epithelium at the other. From these analyses it became clear that with progression of the lesion the immunostaining patterns became more complex, not only when analyzing intensity differences, but also with respect to the distribution of in particular 5-hmC positive cells throughout the individual lesions.

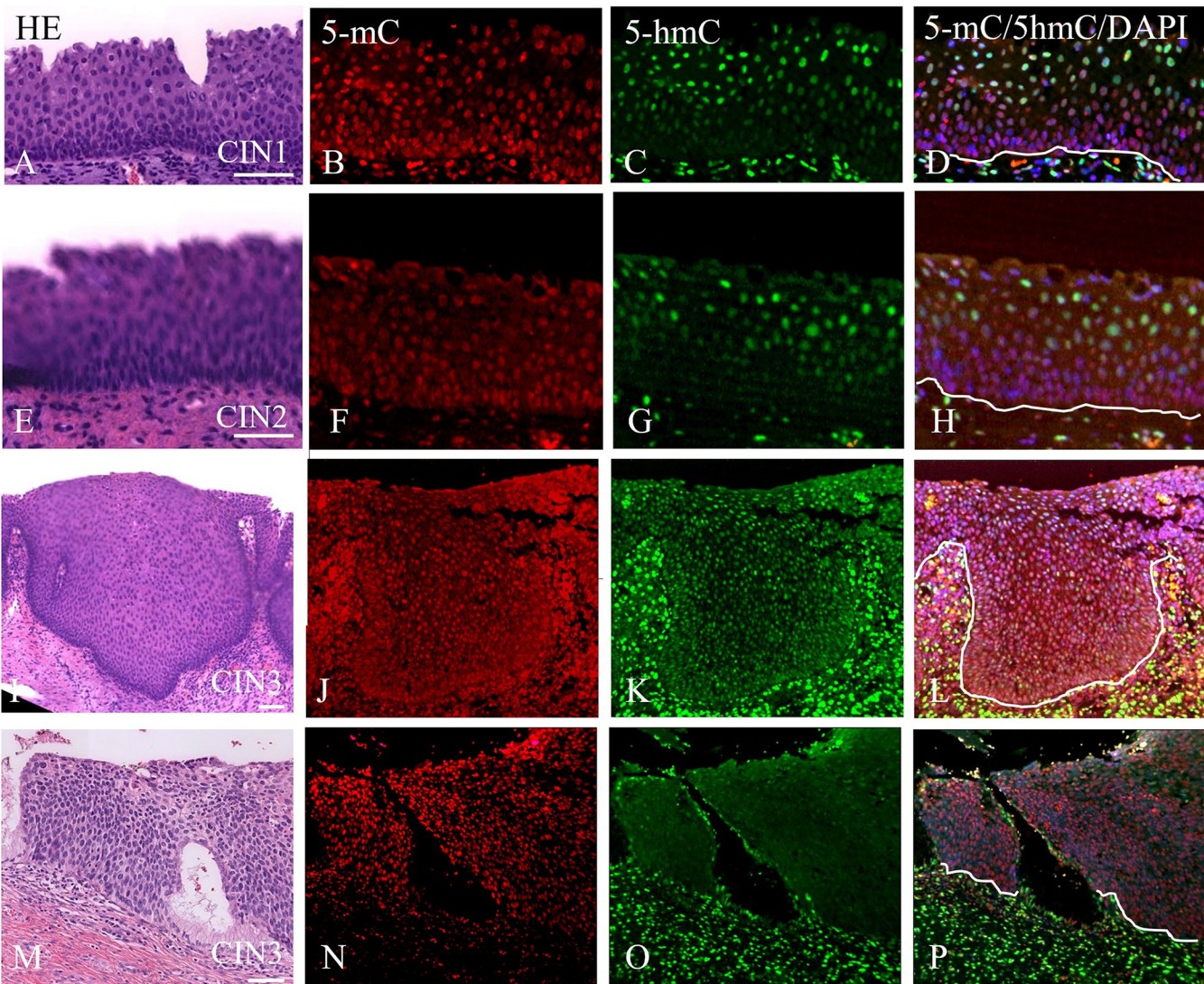

**Fig 6. Complex immunofluorescence staining patterns for 5-mC and 5-hmC in CIN lesions. A-D)** CIN1 area; **E-H)** CIN2 area and **I-P)** CIN3 areas. **A, E, I and M)** H&E stainings; **B, F, J and N)** 5-mC; **C, G, K and O)** 5-hmC; **D, H, L and P)** merged images of 5-mC and 5-hmC with DAPI counterstaining. Images were collected at lower magnification with a 20 times objective. Note the different staining patterns for 5-hmC with a variation in the basal/parabasal layers being strongly positive to weakly positive. Scale bar in **A** indicates 100 μm (same magnifications in **B-P**). The lines in **D, H, L** and **P** mark the position of the basement membrane. Additional images collected at higher magnification and depicting the immunostaining patterns of the lower compartments of the epithelium in more detail are illustrated in S2 Fig for different CIN lesions.

Fig 6 summarizes different epithelial immunofluorescence distribution patterns that were recognized for 5-mC and 5-hmC in areas marked by the pathologist as CIN1, CIN2 and CIN3. We want to stress that CIN1 lesions exhibited similar patterns for 5-mC and 5-hmC as recorded for the normal squamous epithelium (Fig 6A–6D; see also S2 Fig), with intense (3 out of 8 areas analyzed) to weak (5 out of 8 analyzed) staining in the basal compartment and positive immunostaining in the intermediate and superficial layers. On top of these distributions both 5-mC and 5-hmC positive nuclei were intermingled with weakly stained and negative nuclei recognized in 5 out of 8 CIN1 areas analyzed. These differences were most pronounced for 5-hmC as compared to 5-mC. The distribution pattern as shown for CIN1 in Fig 6A–6D, exhibiting the low intensity for 5-hmC in the basal compartment, was frequently

observed in patients diagnosed with a higher CIN grade, as illustrated in Fig 6E–6H, showing weak staining in the lower compartment, but a higher reactivity in the upper layers. This pattern was recognized in 30 out of 84 areas graded higher than CIN1 (see Table 1: the 84 areas comprised 55 CIN3, 7 solitary CIN2 and 22 coinciding CIN2 areas in CIN3).

The CIN2 and CIN3 areas showed also for both 5-mC and 5-hmC immunostaining throughout the entire epithelium, next to the low intensity staining for 5-hmC in the lower compartment of the epithelium (Fig 6I and 6L; 34 out of 84 areas). Finally, in about 20% of the patients diagnosed with CIN3 (5 out of 25 patients), a (very) weak staining for 5-hmC was observed throughout the entire epithelium (Fig 6O), while most nuclei were strongly positive for 5-mC (Fig 6N). Quantitative data were obtained to support the different distribution patterns as illustrated in Figs 1 and 6 (and S2 Fig). As shown in S3 Fig these quantitative analyses support the semiquantitative observations that largely three different 5-hmC distribution patterns can be distinguished in CIN lesions: 1) a pattern with a high intensity staining for 5-hmC in the basal cell layer, a reduced intensity in the parabasal cell layer and increasing intensity in the direction of differentiation of the epithelium to the more upper layers of the epithelium seen in CIN1; 2) a pattern with a low intensity staining for 5-hmC in the lower compartment of the epithelium including the bc, pbc layers and lower compartment of the icl seen in CIN1-3; 3) a complex staining pattern for 5-mC and 5-hmC throughout the epithelium, with a strong staining at the bc layer for both 5-mC and 5-hmC seen in CIN2-3.

In summary, CIN lesions showed a heterogeneous staining pattern for 5-hmC. When studying CIN3 cases in more detail, combinations of the patterns described above were found next to each other (20 out of 55 areas; see also S4 Fig for examples). Also, when performing confocal microscopy, CIN lesions were found to show areas where 5-mC and 5-hmC immunostainings were mutually exclusive, with nuclei exhibiting intense staining for 5-hmC, while being negative for 5-mC and vice versa (see also S5 Fig).

## Immunostaining results of 5-mC and 5-hmC in premalignant endocervical glandular lesions

Cervical adenocarcinoma in situ (AIS) lesions, previously analyzed for SOX17 gene expression and methylation [26] showed a strong immunostaining for 5-mC in all glandular structures, both normal and neoplastic. Strikingly, compared to the normal glandular structures, a drastic decrease in 5-hmC immunostaining was observed in nearly all regions with AIS, which were recognized by p16 positivity and partial SOX17 negativity (Fig 7). Small areas of normal (p16 negative, SOX17 positive) glandular epithelium inside the AIS did, however, show the typical, strong 5-hmC immunoreactivity.

Higher magnification images in the separated colors as depicted in Fig 8 illustrate the differences in staining intensity for 5-mC and 5-hmC in more detail. When studying AIS lesions from different patients some variation is seen for 5-hmC staining intensity, but in all 12 AIS cases a drastic decrease of hydroxymethylation was obvious. Also, when using TE and pepsin/HCl as antigen retrieval methods, to exclude that the observed reduction of staining for 5-hmC resulted from increased DNA compactness of the 5-hmC domains, such a decrease was still clearly visible (data not shown).

## Immunostaining results for 5-mC and 5-hmC in cervical adenocarcinomas

Also, in nearly all 14 adenocarcinoma cases studied a dominant fluorescence for 5-mC was observed, while 12 out of 14 patients showed a greatly reduced immunostaining for 5-hmC, indicating a strong reduction of hydroxymethylation in cervical adenocarcinomas. Fig 9A–9C illustrate this phenomenon at a low magnification, but a decrease of hydroxymethylation is

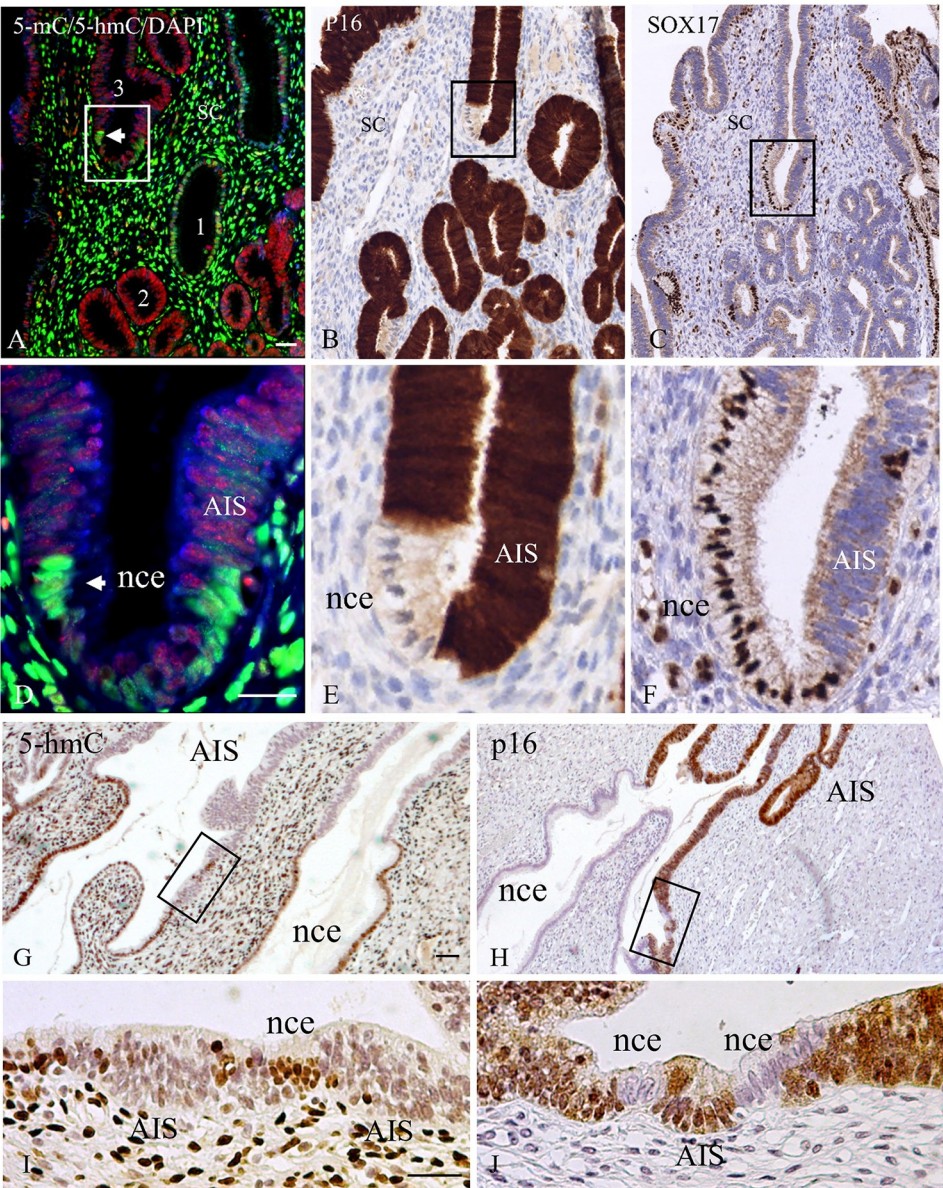

**Fig 7. Immunostaining patterns of 5-mC and 5-hmC in normal glandular epithelium and adenocarcinoma in situ (AIS).** Merged image of 5-mC in red and 5-hmC in green with blue DNA counterstaining using DAPI. Note the dramatic decrease in 5-hmC immunofluorescence in the AIS structures (gland 1 normal columnar epithelium, glands 2 and 3 containing AIS structures). The arrows in **A** and **B** point to normal columnar epithelium (nce) adjacent to AIS cells, the latter being identified by p16 positivity (**B**) and SOX17 positivity (**C**). **D**, **E** and **F**, higher magnifications of boxed areas in **A**, **B** and **C**. Normal columnar epithelium is characterized by the absence of immunostaining for p16 and SOX17. Note that the images in **A**, **B** and **C**, although being derived from the same area in the AIS lesion, do not represent consecutive sections, resulting in slight differences in cellular distribution patterns. **G-J**) Brightfield images of 5-hmC immunostaining patterns of a cervical AIS, showing mutual exclusive staining for 5-hmC and p16 in consecutive tissue sections. **G** and **I**) show 5-hmC positivity only in the normal columnar epithelium (nce); **H** and **J**) show p16 positivity only in the AIS region. **I** and **J**, higher magnifications of boxed areas in **G** and **H**. Scale bars indicate 30 μm.

even more evident at higher magnification (Fig 9D and 9E). 5-mC levels were not altered in all adenocarcinomas when comparing fluorescence intensities in normal columnar, malignant and stromal nuclei. In a minor fraction of the adenocarcinoma cases (2 out of 14), however, an

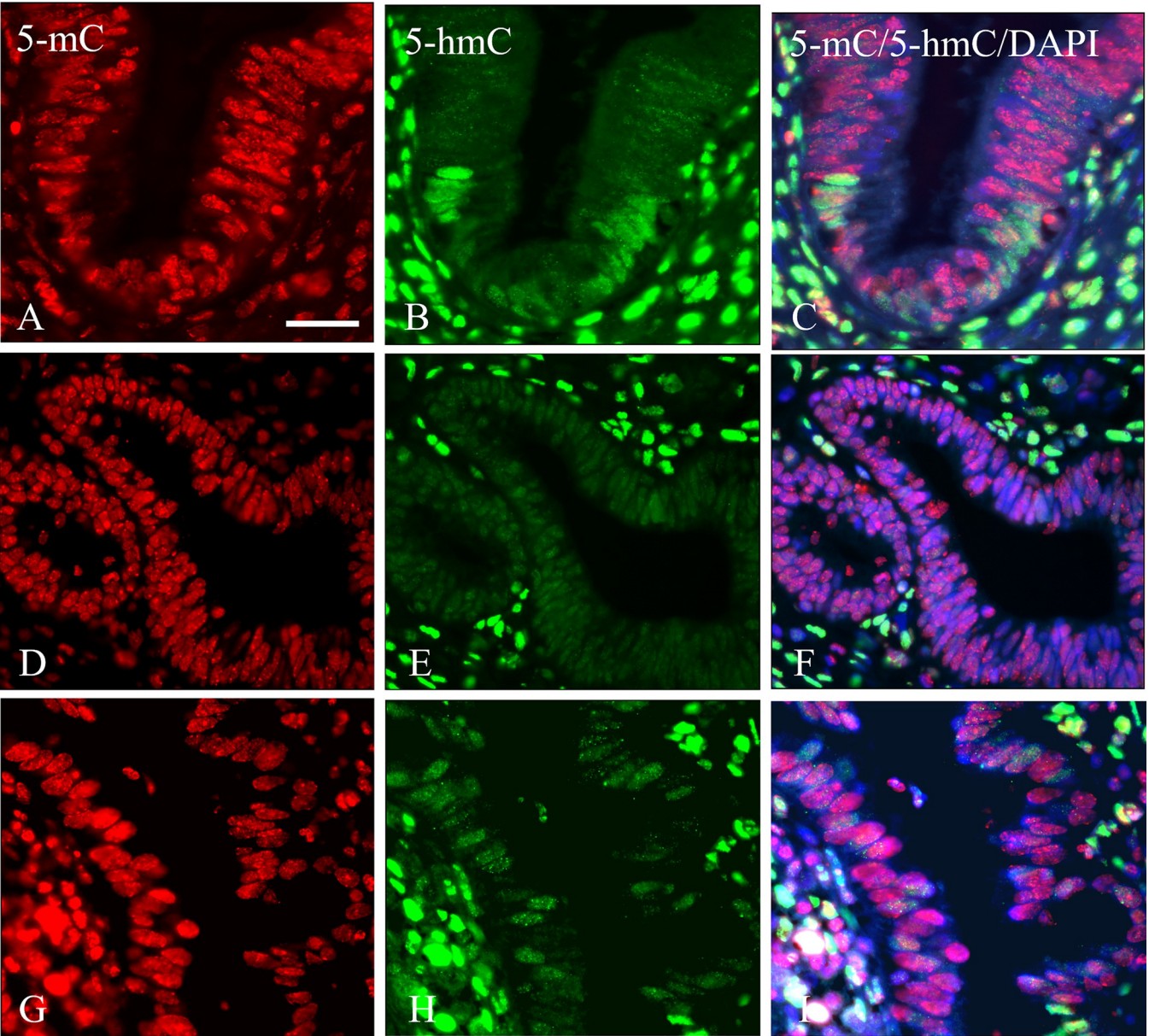

**Fig 8. Immunostaining of 5-mC and 5-hmC in adenocarcinoma in situ (AIS). A**, **D** and **G**) 5-mC; **B**, **E** and **H**) 5-hmC; **C**, **F** and **I**) merged images of 5-mC and 5-hmC with DAPI counterstaining, respectively. Note the weak intensity of 5-hmC in the AIS structures. Scale bar in **A** indicates 20 μm (same magnifications in **B**-**I**).

intense staining for 5-hmC was seen (Fig 9H and 9K), while 3 out of 14 adenocarcinoma cases showed heterogeneity for the loss of hydroxymethylation levels (See S6 Fig).

## Discussion

The transformation zone of the uterine cervix is an ideal tissue to study the stepwise process of carcinogenesis because subsequent stages of metaplastic, dysplastic and (micro)invasive changes can be easily accessed and molecular switches that take place during this process can be studied in a histo-morphological context. Triggered by the many gene specific methylation

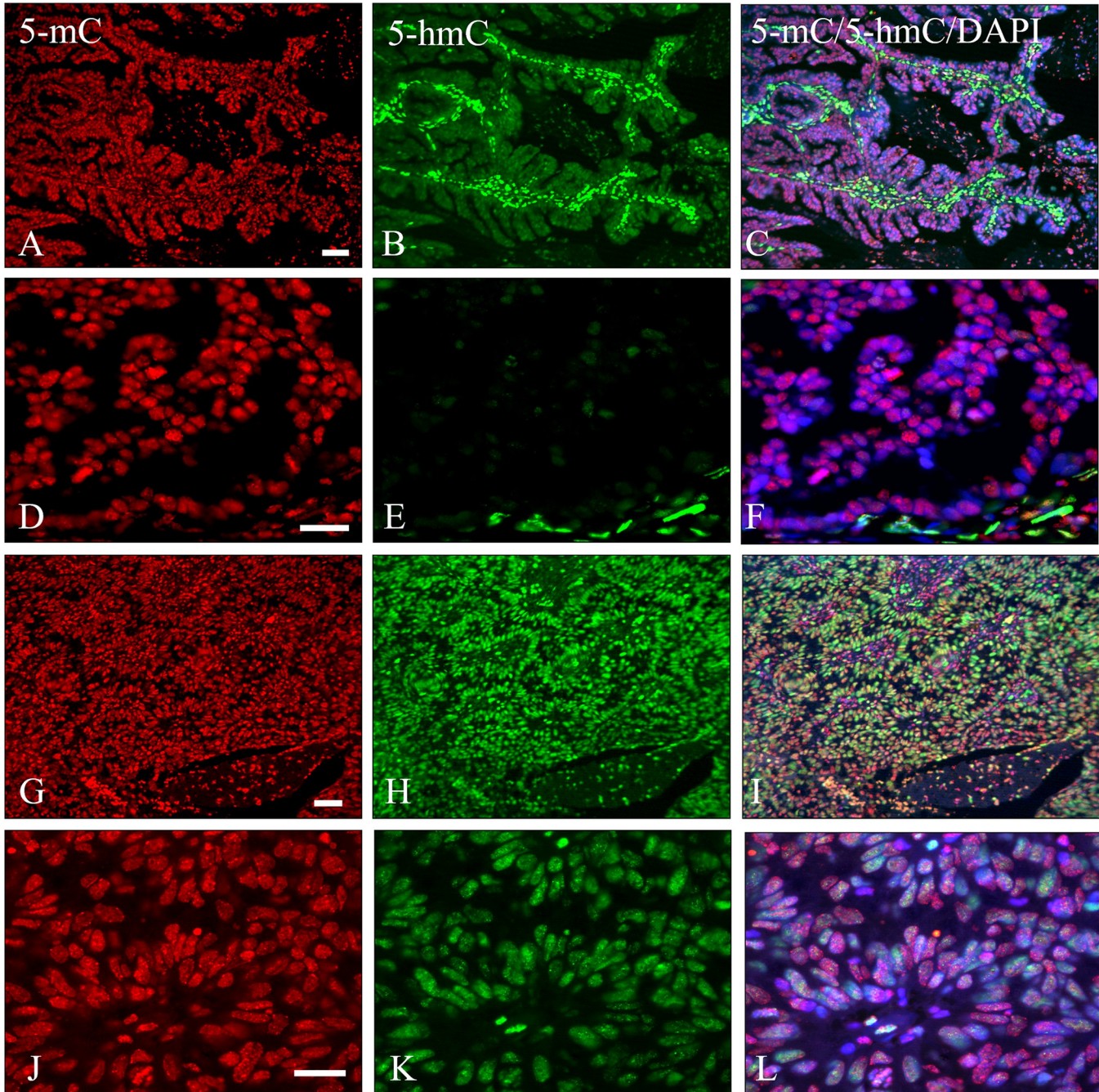

**Fig 9. Immunostaining results of 5-mC and 5-hmC in adenocarcinomas. A**, **D**, **G** and **J**) 5-mC; **B**, **E**, **H** and **K**) 5-hmC; **C**, **F**, **I** and **L**) merged images of 5-mC and 5-hmC with DAPI counterstaining. Note the weak/negative staining for 5-hmC in **B** and **E** and intense staining in **H** and **K**. Scale bars indicate 100 μm in **A** and **G** (same magnifications in **B**, **C**, **H** and **I**) and 20 μm in **D** and **J** (same magnifications in **E**, **F**, **K** and **L**).

studies in cervical lesions (see Introduction) we decided to study the process of (hydroxy) methylation at a more global scale in the normal and (pre)malignant lesions using an immunohistochemical approach. Contradictory results exist in the literature concerning the methylation and hydroxymethylation status of the genome in cervical cancer. Several authors state that the methylation level of cytosine (5-mC) is reduced in cervical cancers when compared to

normal epithelium [11, 12], while others provide evidence for an increase of 5-mC [27]. Downregulation of 5-hmC was shown to be associated with the progression of CIN lesions [28], with 5-hmC being significantly decreased in cervical squamous cell carcinoma as compared to CIN lesions and normal cervical tissues [11, 29]. Finally, most of the studies into gene methylation and hydroxymethylation in the cervix have been performed with ectocervical squamous lesions, while endocervical glandular lesions, such as AIS and adenocarcinoma, have not been studied to a great extent. In the underlying study we show that the global 5-hmC levels decrease to a certain extent with progression of CIN, but that particularly AIS and adenocarcinoma lesions exhibit a steep decrease in global hydroxymethylation, while 5-mC remained largely detectable in these (pre)malignancies to a similar extent as in their normal counterparts. The major advantage of using an immunohistochemical approach, as opposed to the more integrated approaches used by others, is that we obtained spatially resolved information that allowed us to correlate the epigenetic switches to the position of individual cells in morphologically heterogeneous tissue samples. The observed (hydroxy) methylation patterns in the normal epithelia and their progenitor cells provide clues about their relation and differentiation status, while the differences in the hydroxymethylation process between squamous and glandular uterine cervical lesions can partially be explained in light of mechanisms that characterize the different carcinogenic pathways, including the physical status of HPV, genomic alterations and instability, gene-specific methylation profiles and the activity of TET proteins.

## Dynamics in 5-mC levels

Only minimal differences in fluorescence intensity for 5-mC were observed when comparing normal (including metaplastic) squamous epithelium, normal glandular epithelium and (pre) neoplastic lesions. A high degree of heterogeneity in staining intensity was, however, observed in the different histological compartments, with intensely and weakly stained nuclei intermingled and significant differences in immunofluorescence intensity between these nuclei. This heterogeneous staining pattern was not only seen throughout the basal/parabasal, intermediate and superficial compartments of the squamous epithelium, also the normal columnar epithelium with underlying reserve cells showed intermingled intensely and weakly stained nuclei in the different stretches of this single-layered epithelium.

The observed random distribution of intensely and weakly stained nuclei in the normal tissues follows a pattern similar to the random distribution of maternal or paternal inactivation of the X chromosome as a result of DNA methylation [30]. This X chromosome inactivation occurs early during embryo development and is maintained during life. It remains to be seen whether or not a correlation exists between the process of X chromosome inactivation and the heterogeneity of 5-mC immunoreactivity.

The intermingled staining pattern as observed in the premalignant lesions, and the observation that nuclei with intense or weak staining were not clustered, is unexpected based on the assumption that most of these lesions result from clonal expansion. Our data may, however, be in line with those of Ueda et al. (2003), who showed that most CIN lesions are monoclonal in composition, but that CIN cases in which the HPV genome was present in episomal form were polyclonal [31]. This may imply that HPV viral integration into the host genomic DNA is associated with progression from a polyclonal to monoclonal status in CIN. As a result, also the 5-mC staining patterns may vary in such CIN lesions based on the physical status of the virus.

Several authors have described changing levels of 5-mC during carcinogenesis, but their results are interpreted differently. Zhang et al. (2016) reported a significant difference in staining intensities when comparing normal to cancer samples [29]. However, since the presented

images showed no staining for 5-mC in the stromal compartment, these data should be interpreted with care. Wang et al. (2019) visualized 5-mC utilizing brightfield microscopy and showed intensely staining nuclei in both control and cancer samples [27]. In their samples the ratio of intensely stained nuclei intermingled with nuclei showing no staining altered during progression. Kato (2020) showed data suggesting an abrupt loss in 5-mC methylation with progression from CIN2 to CIN3 and cancer [28]. The corresponding plots of immunofluorescence showed, however, a wide range of intensities rather than an abrupt change. Our data align with those of Bhat et al. (2017) who measured a two-fold difference for 5-mC when comparing DNA isolated from normal epithelium and cervical squamous cell carcinoma [11]. This integrated approach does, however, not show the different levels of 5-mC in individual cells and tissue compartments, which did become clearly apparent from our immunohistochemical approach.

## Dynamics in 5-hmC levels

For 5-hmC significant differences in fluorescence intensity were seen in the different compartments of the normal epithelia and even more so during the development of the two types of (pre)neoplasia.

While normal ectocervical squamous epithelium often shows a strong 5-hmC staining in (part of) the basal cells, this compartment is largely negative when located adjacent to the SCJ or within (im)mature metaplastic epithelium. Also, the reserve cells underlying the glandular epithelium, which are the progenitor cells for metaplastic squamous epithelium, show only low levels of 5-hmC or are completely negative. The low level of 5-hmC in reserve cells aligns with the published observations that stem cells or progenitor cells frequently show low levels of 5-hmC [10, 32, 33]. The formation of immature metaplasia results from proliferation of reserve cells [26] and the immature metaplastic epithelium subsequently matures into a squamous epithelium that is indistinguishable from the original ectocervical squamous epithelium. This process occurs in the transformation zone with the new SCJ as the border between squamous and columnar epithelium. In contrast to the basal cells in the original ectocervical squamous epithelium and the columnar epithelium in which 5-hmC levels are high, the parabasal cells in the squamous epithelium showed a lower level of 5-hmC. This difference suggests that these highly proliferative progenitor cells responsible for the renewal of the epithelium (the so-called transient amplifying compartment) have a low level of 5-hmC. An increase in 5-hmC levels was observed during the subsequent process of epithelial differentiation, resulting in relatively high 5-hmC immunofluorescence levels in the terminally differentiated cells in the more superficial layers. These observations align with those of Haffner et al. (2011) who showed that in the basal part of the human adult colon the level of 5-hmC is low, while on top of the epithelium the highest 5-hmC levels were recorded [10]. Also, the cervical columnar epithelium showed a high 5-hmC level, suggesting that these cells can be regarded as terminally differentiated cells with a very low proliferation activity.

When studying CIN lesions, grade-specific 5-hmC immunostaining patterns were recognized. CIN1 lesions showed the highest 5-hmC levels in the basal/parabasal compartment, comparable to the pattern seen in normal squamous epithelium. In contrast, CIN2 lesions often showed an intense 5-hmC immunostaining from basal/parabasal to the upper layers, a pattern shared with CIN3. And finally, part of the CIN3 lesions showed a reduced staining throughout the dysplastic epithelium. A typical pattern for the higher grade CIN lesions was the low 5-hmC level in the basal/parabasal compartment. The grade-specific patterns align with non-disturbed (low-grade CIN) and disturbed differentiation (high grade CIN) of the epithelium upon infection with HPV (see also S1 Fig). In the literature reduced 5-hmC levels

were reported when cancer was compared with normal epithelium, but again data with respect to the spatial distribution within the tissues and levels in the different grades of CIN were lacking so far [11, 27–29].

In the underlying study we report for the first time on the 5-hmC levels in (pre)malignant glandular cervical lesions. In nearly all premalignant glandular (AIS) lesions, as well as adeno-carcinomas, a (very) low 5-hmC level was detected, certainly when compared to the normal columnar epithelium. A straightforward explanation for this finding could be that changes in TET2 activity underly the reduced hydroxymethylation activity. Such reduced expression or activity levels of TET2 and resulting low 5-hmC levels have been reported for many solid tumor types as well as hematopoietic disorders [14, 15, 17, 34–36]. Studies are in progress to correlate the 5-hmC immunostaining patterns to the expression patterns of TET proteins in normal cervical epithelia and the (pre)malignancies derived therefrom.

## Influence of HPV infection and integration on 5-mC and 5-hmC levels

HPV infection of metaplastic epithelium or reserve cells in the TZ can result in a squamous intraepithelial lesion. The viral proteins E6 and E7 are expressed and uncouple cell growth arrest, apoptosis and differentiation of the host cell primarily through the inactivation of p53 and pRb, respectively. Also, HPV E7 binds to DNA (cytosine-5)-methyltransferase 1 (DNMT1) and stimulates its activity, while at the same time transcription of DNMT1 is activated through the pRb/E2F pathway and HPV E6 upregulates DNMT1 by suppression of p53 [37]. This increased DNA methylation activity then (de)regulates the expression of specific genes.

Several studies have shown that the frequency of methylation of candidate genes increases with increasing severity of the cervical lesion, suggesting that these changes already occur early in cancer development. DNA methylation is found to be more common in invasive cervical carcinoma and CIN3 as compared to CIN1/2 [38]. This can to a certain extent be explained by the decreased hydroxymethylation status as detected in the underlying study for the higher grade CIN lesions. Kato et al. (2020) recently showed that 5-hmC levels are decreased between CIN2 and CIN3 through the TP53-A3B pathway [28]. These authors also propose that, since A3B (apolipoprotein B mRNA editing enzyme) could impair genome stability, 5-hmC loss might increase the chances of accumulating mutations and of progressing from CIN3 to cervical cancer.

Besides the locus-specific methylation events, cell cycle deregulation disturbs the coordination between DNA replication and activity of DNA methyltransferases, leading to hypomethylation of repetitive regions in the genome, finally resulting in genomic instability.

Integration of HPV DNA into the host cell genome is a key event in HPV-mediated carcinogenesis [39]. In CIN1 and CIN2 the virus is mostly present in an episomal form which allows viral replication, during which differentiation is moderately altered. Upon integration into the human genome, which occurs relatively late in the development of CIN3, differentiation is interrupted. The AIS lesions examined in the underlying study, however, contained integrated virus in nearly all cases [40]. The cases with low 5-hmC levels in our study therefore align very well with the cases in which the virus was shown to be integrated into the host genome. To explain this direct correlation between the integration of the virus on the one hand and the low 5-hmC levels at the other, we hypothesize this to be the result of TET2 deregulation. The 5-hmC moieties resulting from TET2 activity are used as a tag at stalled replication forks, formed during replication stress in order to recruit enzymes for the base excision repair during the cell cycle. When TET2 is inactive as a result of mutations or not expressed as a result of promotor CpG methylation, the stalled replication forks are not degraded but

stabilized. These conditions contribute to genomic instability [41, 42], in particular within the so-called fragile sites of the genome. McBride and Warburton (2017) hypothesized that HPV replicates its DNA by applying the host DNA response machinery adjacent to areas in the genome that are susceptible to replication stress [43]. These fragile sites align with transcriptionally active regions (enriched for repetitive elements and CpG dinucleotides) and are candidate regions for HPV integration. We hypothesize therefore that deregulated TET2 activity is the cause of accidental viral integration into the host genome, leading to abrogation of the viral life cycle.

**Clinical utility.** Our results seem to indicate that inhibition of global demethylation, that will normally follow upon cytosine hydroxymethylation, is an important epigenetic switch in the development of cervical cancer. As a consequence, not only the methylation status of specific genes can be used as an indicator for the degree of malignant transformation of the cervical squamous and glandular lesions [15–17], but also the detection of global hydroxymethylation by means of immunohistochemical staining of 5-hmC. In case of glandular lesions an inhibition of global demethylation was frequently recognized in AIS, the prestage of adenocarcinoma, and may become of clinical utility in combination the conventional H&E staining. Such a correlation between the loss of 5-hmC levels and the degree of malignant progression was less evident for the squamous premalignant lesions. For that reason, the potential clinical utility of 5-mC and 5-hmC immunostaining is limited for these squamous lesions. In diagnostically challenging clinical CIN samples, however, the 5-hmC staining patterns, showing either positivity throughout the full thickness of the epithelium or a negative epithelium, are indications for a high-grade lesion in addition to the morphological criteria.

When interpreting the result of such an immunohistochemical analysis one has to realize, however, that in addition to 5-hmC, the Tet proteins can generate 5-formylcytosine (5-fC) and 5-carboxylcytosine (5-caC) from 5-mC. A potential alternative 5-mC demethylation mechanism has therefore been suggested by Ito et al [2], in which the Tet proteins oxidize 5-mC not only to 5-hmC, but also to its aldehyde form 5-fC and the carboxylic acid form 5-caC.

## Supporting information

**S1 Fig. Direct correlation of immunostaining results for 5-hmC, HPV, SOX2 and SOX17 in tissue sections of normal squamous epithelium, metaplastic epithelium and CIN3. A**) H&E-stained stretch of squamous epithelium. **B**) Immunostaining of immature metaplastic epithelium by SOX17 and **C**) immunostaining of p16 for HPV-infected CIN3. **D-F**) Higher magnifications of boxed area 3 indicated in **A**, showing SOX2 immunostaining (**D**), HPV detection by chromogenic in situ hybridization (**E**) and 5-hmC immunofluorescence (**F**). Straight lines in **D-F** mark a collision area with the left area being SOX2 negative, HPV negative and 5-hmC weak to negative in the basal/parabasal compartment of the immature metaplastic epithelium. In the right area SOX2 stained positive, HPV is detected and 5-hmC shows an overall weak to negative staining and scattered positive nuclei. The lines in **F-I** indicate the position of basement membrane. **G-I**) Higher magnifications of boxed areas 1, 2 and 4, respectively, showing 5-hmC immunostaining. In the area distal (box 1) from the infected TZ (box 3) the basal cells showed a high fluorescence for 5-hmC (compare Fig 1**D**). In the area in box 4, the dysplastic epithelium showed a weak staining for 5-hmC (compare Fig 6**K**). Scale bars indicate 500 μm in **A** (same magnifications in **B** and **C**), 100 μm in **D** (same magnifications in **E-F**), and 20 μm in **G** (same magnifications in **H** and **I**).
(TIF)

**S2 Fig. Complex immunofluorescence staining patterns for 5-mC and 5-hmC in CIN lesions. A-C**) CIN1 area in CIN1 patient; **D-F**) CIN1 area in CIN3 patient; **G-I**) CIN3 area

diagnosed in CIN3 patient; **J-L**) CIN3 area in CIN3 patient. Images were captured using a 40 times oil objective and are higher magnification images of cases shown in Fig 6. **A**, **D**, **G** and **J**) 5-mC; **B**, **E**, **H** and **K**) 5-hmC; **C**, **F**, **I** and **L**) merged images of 5-mC and 5-hmC with DAPI counterstaining. Note the different staining patterns for 5-hmC with a variation in the basal/ parabasal layers being strongly positive to weakly positive or even negative. Scale bar in **A** indicates 20 μm (same magnifications in **B**-**L**). The lines in **C**, **F**, **I** and **L** mark the position of the basement membrane (bm). The stromal cell layer, basal and parabasal cells and intermediate cell layers are indicated as sc, bc, pbc and icl respectively.
(TIF)

**S3 Fig. Quantitative analysis of 5-mC and 5-hmC immunofluorescence intensity (Arbitrary Units, AU) distributions in normal squamous epithelium, CIN1 and CIN3. A, D,** and **G**) Merged immunofluorescence images of 5-mC (red), 5-hmC (green) and DAPI (blue) in normal squamous epithelium, CIN1 and CIN3. Lines depicted in **A**, **D** and **G** show the immunofluorescence scan track for the three individual colors from the lower to the higher compartment of the epithelium. **B**, **E** and **H**) Intensity plots for DAPI and 5-mC. **C**, **F** and **H** Intensity plots for 5-hmC. **C** shows an example of the pattern with a high intensity staining for 5-hmC in the basal cell layer (bc layer) with a reduced intensity in the parabasal cell layer (pbc layer). The intensity for 5-hmC increases in the direction of differentiation of the epithelium to the more upper layer of the epithelium (intermediate cell layer; icl). **F**) shows an example of the pattern with a low intensity staining for 5-hmC in the lower compartment of the epithelium including the bc, pbc layers and lower compartment of the icl. Compare the difference in fluorescence intensity distribution between 5-mC and 5-hmC. **I**) shows an example of the pattern with a complex staining for 5-mC and 5-hmC throughout the epithelium, with a strong staining at the bc layer for both 5-mC and 5-hmC, and quantitatively supporting the observed inter- and intranuclear differences in immunostaining for 5-mC and 5-hmC. Note that the stromal compartment always shows cells with a high nuclear intensity staining for 5-hmC (see also **Figs 1 and 2**), which are localized underlying the epithelial basal cell compartment (see **S3D Fig**). These stromal cells have not been included in this quantitative analysis (see **S3E** and **S3F Fig**).
(TIF)

**S4 Fig. Immunofluorescence staining patterns for 5-mC and 5-hmC in CIN3 lesions with overlying columnar epithelium, showing extensive heterogeneity for methylation and hydroxymethylation. A-G**) Merged images of 5-mC in red and 5-hmC in green with DAPI counterstaining in blue. Two patient samples are shown, one in **A**-**C** and the other in **D**-**G**. Scale bar in **A** indicates 20 μm (same magnifications in **B**-**G**). The lines mark the position of the basement membrane. In **A** and **D** arrowheads point to normal columnar epithelium and columnar epithelium on top of the dysplastic squamous epithelium. In **C** and **E** +: indicates areas with an increased 5-hmC immunostaining as compared to * indicating areas with a decrease of 5-hmC immunostaining.
(TIF)

**S5 Fig. Confocal images of double-label immunofluorescence staining patterns for 5-mC and 5-hmC in a CIN3 lesion. A**) Merged image of 5-mC in red and 5-hmC in green with blue DAPI counterstaining; **B**) Higher magnification of boxed area in **A**; **C**) 5-mC with blue DAPI counterstaining. **D**) 5-hmC with blue DAPI counterstaining. Note the mutually exclusive staining patterns for 5-mC and 5-hmC. Nuclei with dominant 5-hmC staining indicated with 1 and 5-hmC dominant staining indicated with 2. Scale bar in **A** indicates 10 μm and 20 μm in **B** (same magnifications in **C** and **D**).
(TIF)

**S6 Fig. Immunofluorescence staining results for 5-mC and 5-hmC showing heterogeneity for 5-hmC in cervical adenocarcinoma. A**) H&E-stained dysplastic glands in adenocarcinoma. **B**) 5-mC; **C**) 5-hmC; **D**) merged mages of 5-mC and 5-hmC with DAPI counterstaining. Scale bar in **A** indicates 100 μm (same magnifications in **B**-**D**).
(TIF)

**S1 Table.**  **A** Antibody characteristics and optimized detection methods applied for the single- and double-label immunofluorescence analyses of 5-mC and 5-hmC. **B** Antibody characteristics and optimized detection methods applied for the bright field immunohistochemical analyses.
(PDF)

## Acknowledgments

We thank Dr. Jack Cleutjens (Dept of Pathology, Maastricht University Medical Center, Maastricht, The Netherlands) for help with scanning of the tissue slides. We are grateful to Frans Kwaspen (PanPath, Budel, The Netherlands) for providing HPV probes for in situ hybridization.We would also like to acknowledge the input of Prof. Dr. Leendert Looijenga (Princess Máxima Center for Pediatric Oncology, Utrecht, The Netherlands) in the initial set up of the experiments.

## Author Contributions

**Conceptualization:** Jobran M. Moshi, Frank Smedts, Frans C. S. Ramaekers, Anton H. N. Hopman.

**Investigation:** Jobran M. Moshi, Monique Ummelen, Frank Smedts, Frans C. S. Ramaekers, Anton H. N. Hopman.

**Methodology:** Jobran M. Moshi, Monique Ummelen, Frank Smedts, Anton H. N. Hopman.

**Supervision:** Frans C. S. Ramaekers, Anton H. N. Hopman.

**Visualization:** Jobran M. Moshi, Monique Ummelen, Anton H. N. Hopman.

**Writing – original draft:** Jobran M. Moshi, Frans C. S. Ramaekers, Anton H. N. Hopman.

**Writing – review & editing:** Frank Smedts.

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
