## [Decision Letter · Decision Letter 0]

28 Jun 2023

PONE-D-23-13996Inhibition of cytosine 5-hydroxymethylation during progression of cancer precursor lesions in the uterine cervixPLOS ONE

Dear Dr. Hopman,

Thank you for submitting your manuscript to PLOS ONE. After careful consideration, we feel that it has merit but does not fully meet PLOS ONE’s publication criteria as it currently stands. Therefore, we invite you to submit a revised version of the manuscript that addresses the points raised during the review process.

We look forward to receiving your revised manuscript.

Kind regards,

Ricardo Ney Oliveira Cobucci, Ph.D

Academic Editor

PLOS ONE

“Jobran Moshi: His work was supported by a grant from Jazan University, Jazan, Saudi Arabia (grant number 1067568103). This study was part of the PhD study of Jobran Moshi.”

Reviewers' comments:

Reviewer's Responses to Questions

**Comments to the Author**

1. Is the manuscript technically sound, and do the data support the conclusions?

Reviewer #1: Yes

Reviewer #2: Yes

Reviewer #3: No

2. Has the statistical analysis been performed appropriately and rigorously? 

Reviewer #1: Yes

Reviewer #2: N/A

Reviewer #3: No

3. Have the authors made all data underlying the findings in their manuscript fully available?

Reviewer #1: Yes

Reviewer #2: Yes

Reviewer #3: Yes

4. Is the manuscript presented in an intelligible fashion and written in standard English?

Reviewer #1: Yes

Reviewer #2: Yes

Reviewer #3: Yes

5. Review Comments to the Author

Reviewer #1: Jobran et al. used the citrate pH 6.0 retrieval protocols to detect the levels of 5-mC and 5-hmC in FFPE tissues from normal cervix, precervical lesions, and cervical cancer patients. They concluded that the inhibition of demethylation, which normally follows cytosine hydroxymethylation, is an important epigenetic switch in the development of cervical cancer. In principle, this is a very interesting piece of work. Their findings provide new evidence for understanding the mechanisms of cervical disease progression. However, before the article can be formally published, there are still several questions that need to be explained by the authors.

1.“Figure 6. Immunofluorescence staining patterns for 5-mC and 5-hmC in CIN lesions”. Please provide distribution maps of CIN-associated mild, moderate, or severe atypical hyperplasia cells and cancer cells in situ, and explain whether the levels of 5-mC and 5-hmC are expressed in these abnormal cells？

2."Figure 7: Immunostaining patterns of 5-mC and 5-hmC in normal glandular epithelium and adenocarcinoma in situ (AIS)." (1) Please provide distribution maps of the cells of adenocarcinoma in situ (AIS), and explain whether the levels of 5-mC and 5-hmC are expressed in these abnormal cells？ (2) In addition, the three images (A-B-C) in Figure 7 seem to display different tissues rather than consecutive sections of the same tissue. It is unclear whether these images represent the same region or area within the tissue. The authors should provide a clear explanation regarding the different of these images.

3.Figure 6, it appears that the labels D-F should represent CIN2 rather than CIN1-3, as indicated in the paper text. Figure 7, the Line 4 of the figure legend mentions ‘E’, but there is no corresponding E figure in Figure 7. Please clarify and correct this inconsistency in the figures.

4.In the "Clinical utility" section of the Discussion, "Our results seem to indicate that inhibition of global demethylation, which will normally follow upon cytosine hydroxymethylation, is an important epigenetic switch in the development of cervical cancer.” It is well known that the demethylation process is complex and involves multiple stages, including hydroxymethylation, aldehyde methylation, and carboxymethylation. Therefore, the authors should provide additional explanations about the view point.

5.To evaluate differences in immunofluorescence staining results, it is recommended to use quantitative or semi-quantitative methods such as image analysis software instead of relying on visual inspection. Quantitative methods provide a more objective and reliable analysis of stain strength.

Reviewer #2: Numerous genomic modifications, including epigenetic changes that result in abnormal gene expression as well as altered DNA sequences, have an impact on cancers progression. The DNA methylation and demethylation have recently received a lot of interest. Together with the traditional histo-pathological characteristics, epigenetic profiles may improve the clinical treatment of patients.

The manuscript by Moshi et al. discusses the potential prognostic significance of 5-cytosine epigenetic modifications for progression of cancer precursor lesions in the uterine cervix. The authors discovered a substantial association between the distribution and expression of 5-mC and 5-hmC and the pathologic stages of cervical cancers.

The manuscript is generally well-written and interesting to the field. High-quality figures that clearly show the spatial distribution of 5-mC and 5-hmC across cervical carcinoma grading are its strength. The article generally fulfills the requirements for publication in PLOS ONE.

Please find below some of my concerns:

Introduction

The introduction provided background for the concepts and discussed the study objectives. However, in the sentence: „Usually, during carcinogenesis, the methylation level around specific genes is increased” (page 3), it would be helpful to include the examples of aforementioned genes as well as to clarify whether they are associated to any particular types of cancer or are more general.

Materials and Methods

This section of the manuscript was well prepared, including multiple details.

The points for revision:

- It was not specified how many healthy controls were examined, despite the experiments involving 65 cervical patients. (page 5)

- Since the text and the table exhibited differences (65 vs. 60 respectively), the number of cervical samples needs to be clarified. (page 5)

- The cervical grade abbreviations in Table 1 would be easier to understand if they were explained within the legend. (page 5)

- In my opinion, it was not required to reconsider the effects of antigen retrieval protocols for the discovery of epigenetic DNA modifications. The authors have previously stated the benefit of citrate epitope unmasking in preserving cell nucleus morphology during immunofluorescence detection of 5-mC and 5-hmC (Moshi et al., Histochem Cell Biol 2023). (page 5-6)

- Ensure that the reagent names are accurate, please double-check them, e.g., buffer PBT or PBST (page 6)

Results

The points for revision:

- As it was discussed above, authors have previously stated the benefit of citrate epitope unmasking in preserving cell nucleus morphology during immunofluorescence detection of 5-mC and 5-hmC (Moshi et al., Histochem Cell Biol 2023).

Therefore, I do not think it is essential to include this chapter and supplemental figure S1 (chapter: Choice for the citrate pH 6.0 retrieval protocol for the detection of 5-mC and 5-hmC in FFPE tissue sections. – page 8, Supplemental Figure S1 – page 29) as well as the text: “Also, when using TE and pepsin/HCl as antigen … was still clearly visible” and supplemental figure S4 (chapter: Immunostaining results of 5-mC and 5-hmC in premalignant endocervical glandular lesions – page 18, Supplemental Figure S4 – page 32) in the manuscript.

- The headline of the chapter:“5-mC and 5-hmC immunostaining levels in normal ectocervical squamous epithelium (Figures 1-3)” is not accurate since authors included information about the metaplastic epithelium. The benign condition known as squamous metaplasia of uterine cervix presents no risk of malignancy. However, it cannot be defined as a normal state. (chapter: 5-mC and 5-hmC immunostaining levels in normal ectocervical squamous epithelium (Figures 1-3) – page 8)

- The squamous epithelium is formed by multiple layers, such as: basal, parabasal, intermediate and superficial. To make the results more readable, I would consider marking the layer distribution within stained tissue. (Figure 1 – page 9-10)

- The results for 5-hmC expression in Figures 1D and 1G are confusing. Similar layer organization and, hence, corresponding 5-hmC labeling patterns should be found in the squamous epithelium represented by boxes 1 and 2. (Figure 1 – page 9-10)

- What was the rationale behind selecting and presenting samples from four separate patients. (Figure 4 – page 13)

- I would suggest adding arrows to indicate where 5-mC and 5-hmC signals are seen/ not seen within reserve cells. (Figure G, H, J – page 14)

- The signatures of 5-mC/5-hmC/DAPI are missing within Figure 7A. (Figure 7A – page 17)

- The headlines in the results chapters might be altered to eliminate the phrase “figure”. (page 8-18)

Reviewer #3: The Authors of the study performed image analysis of DNA methylation/demethylation in cervical cancer FFPE tissues, using immunostymulator method for 5mC and 5hmC detection, and confocal microscopy for visualization. The study comprised samples from 60 patients.

The major concern of this study is lack of any statistical analysis. The Authors just made a simle comparison of localization of the epigenetic changes in the selected samples, without any quantitative analysis. The pictures are nice, but they lack significance without showing results of the real data analysis (retrieved possibly from optical density values).

6. PLOS authors have the option to publish the peer review history of their article (what does this mean?). If published, this will include your full peer review and any attached files.

Reviewer #1: No

Reviewer #2: No

Reviewer #3: No

---

## [Author Response · Author response to Decision Letter 0]

31 Aug 2023

Reply to suggestions and comments of Reviewer #1:

1. “Figure 6. Immunofluorescence staining patterns for 5-mC and 5-hmC in CIN lesions”. Please provide distribution maps of CIN-associated mild, moderate, or severe atypical hyperplasia cells and cancer cells in situ, and explain whether the levels of 5-mC and 5-hmC are expressed in these abnormal cells？

Reply: Because of the complexity and heterogeneity of the premalignant lesions, particularly in the higher-grade lesions, which contain often combinations of areas with different grades of preneoplasia (CIN), it is difficult to provide a simple correlation between the degree of progression of the CIN lesions on the one hand and the immunostaining results for 5-hmC on the other. For this reason we have included several examples of such complex combinations of immunostaining results and different patterns in the spectrum of CIN lesions. For clarification we have amended the text and legend (see below), and explain the staining patterns in a more general fashion. 

Complex immunostaining patterns for 5-mC and 5-hmC in squamous intraepithelial lesions 

The patient diagnosis as shown in Table 1 was based on the highest histological CIN grade observed in the tissue sample, but virtually all samples contained areas with different grades of CIN. This was especially obvious in the higher-grade lesions. Using H&E staining of the tissue sections of the different patient samples as a guide to recognize specific areas with different grades of CIN, a correlation between the CIN stages with the different immunohistochemical (hydroxy)methylation patterns was made. From these analyses it became clear that with progression of the lesion the immunostaining patterns became more complex. Fig 6 summarizes these different immunostaining patterns recognized for 5-mC and 5-hmC in areas marked by the pathologist as CIN1, CIN2 and CIN3. In the CIN1 lesions we observed similar patterns for 5-mC and 5-hmC as recorded in the normal squamous epithelium (Fig 6A-C), with intense to weak staining in the basal/parabasal compartment and lesser staining in the intermediate and superficial layers. Both 5-mC and 5-hmC positive nuclei intermingled with weakly stained and negative nuclei. A pattern that was observed in nearly all higher grade CIN2-3 lesions, several of which also contained areas with a CIN1 morphology, is illustrated in Figs 6D-F, exhibiting weak to negative staining in the lower compartment, but more reactivity in the upper layers. Most CIN3 areas showed this pattern with a weak staining in the lower compartment of the epithelium, but areas diagnosed as CIN2/3 or CIN3 occasionally showed overlapping immunostaining for both 5-mC and 5-hmC in all epithelial layers (Figs 6 G and H). Finally, in about 20% of the patients diagnosed with CIN3 (5 out of 25 patients), a (very) weak staining for 5-hmC was observed throughout the entire epithelium (Fig 6K), while most nuclei were positive for 5-mC (Fig 6J). In summary, CIN lesions showed a heterogeneous staining pattern for 5-hmC. When studying CIN3 cases in more detail, combinations of the patterns described above were found next to each other (see S2 Fig for examples). Also, when performing confocal microscopy, CIN lesions were found to show areas where 5-mC and 5-hmC immunostainings were mutually exclusive, with nuclei exhibiting intense staining for 5-hmC, while being negative for 5-mC and vice versa (see S3 Fig). 

Fig 6. Complex immunofluorescence staining patterns for 5-mC and 5-hmC in CIN lesions 

A-C) CIN1 area in CIN1 patient ; D-F) CIN1 area in CIN3 patient; G-I) CIN3 area diagnosed in CIN3 patient ; J-L) CIN3 area in CIN3 patient. A, D, G and J) 5-mC; B, E, H and K) 5-hmC; C, F, I and L) merged images of 5-mC and 5-hmC with DAPI counterstaining. Note the different staining patterns for 5-hmC with a variation in the basal/parabasal layers being strongly positive to weakly positive or even negative. Scale bar in A indicates 20 μm (same magnifications in B-L). The lines in C, F, I and L mark the position of the basement membrane (bm). The stromal cell layer, basal and parabasal cells and intermediate cell layers are indicated as sc, bc, pbc and icl respectively. The superficial cell layer is not shown in the fluorescence images. 

2. "Figure 7: Immunostaining patterns of 5-mC and 5-hmC in normal glandular epithelium and adenocarcinoma in situ (AIS)." (1) Please provide distribution maps of the cells of adenocarcinoma in situ (AIS) and explain whether the levels of 5-mC and 5-hmC are expressed in these abnormal cells. In addition, the three images (A-B-C) in Figure 7 seem to display different tissues rather than consecutive sections of the same tissue. It is unclear whether these images represent the same region or area within the tissue. The authors should provide a clear explanation regarding the different of these images.

Reply: We have modified the Figure 7 and changed the legend by indicating the different cellular structures to help reading the figure. Furthermore, we have performed new immunohistochemical stainings in bright field microscopy of consecutive sections to show the direct correlation between the downregulation of 5-hmC in p16 positive areas of AIS. We have included additional images of these results in Figs 7G-J.

Fig 7. Immunostaining patterns of 5-mC and 5-hmC in normal glandular epithelium and adenocarcinoma in situ (AIS) 

A) Merged image of 5-mC in red and 5-hmC in green with blue DNA counterstaining using DAPI. Note the dramatic decrease in 5-hmC immunofluorescence in the AIS structures (gland 1 normal columnar epithelium, glands 2 and 3 containing AIS structures). The arrows in A and B point to normal columnar epithelium (nce) adjacent to AIS cells, the latter being identified by p16 positivity (B) and SOX17 positivity (C). D, E and F, higher magnifications of boxed areas in A, B and C. Normal columnar epithelium is characterized by the absence of immunostaining for p16 and SOX17. Note that the images in A, B and C, although being derived from the same area in the AIS lesion, do not represent consecutive sections, resulting in slight differences in cellular distribution patterns. G-J) Brightfield images of 5-hmC immunostaining patterns of a cervical AIS, showing mutual exclusive staining for 5-hmC and p16 in same area in the AIS lesion (no consecutive tissue sections). G and I) show 5-hmC positivity only in the normal columnar epithelium (nce); H and J) show p16 positivity only in the AIS region. I and J, higher magnifications of boxed areas in G and H. Note in I and J an alternating area with staining patterns for nce and ACIS. Compare with panels A and B depicting normal columnar epithelial cells trapped in ACIS area. Scale bars indicate 30 μm. 

3. Figure 6, it appears that the labels D-F should represent CIN2 rather than CIN1-3, as indicated in the paper text. Figure 7, the Line 4 of the figure legend mentions ‘E’, but there is no corresponding E figure in Figure 7. Please clarify and correct this inconsistency in the figures.

Reply: We have modified the text in the legends of Figs 6 and 7 to be consistent with the paper text. In the Results section, we have changed the text and explained that the Figure illustrates patterns recognized in the different preneoplastic regions; the areas shown in the panels have a single histological CIN characteristic as indicated by the pathologist. 

Fig 6. Complex immunofluorescence staining patterns for 5-mC and 5-hmC in CIN lesions 

A-C) CIN1 area in CIN1 patient ; D-F) CIN1 area in CIN3 patient; G-I) CIN3 area diagnosed in CIN3 patient ; J-L) CIN3 area in CIN3 patient. A, D, G and J) 5-mC; B, E, H and K) 5-hmC; C, F, I and L) merged images of 5-mC and 5-hmC with DAPI counterstaining. Note the different staining patterns for 5-hmC with a variation in the basal/parabasal layers being strongly positive to weakly positive or even negative. Scale bar in A indicates 20 μm (same magnifications in B-L). The lines in C, F, I and L mark the position of the basement membrane (bm). The stromal cell layer, basal and parabasal cells and intermediate cell layers are indicated as sc, bc, pbc and icl respectively. The superficial cell layer is not shown in the fluorescence images. 

Figure 7: We corrected the legend: E should be C.

4. In the "Clinical utility" section of the Discussion, "Our results seem to indicate that inhibition of global demethylation, which will normally follow upon cytosine hydroxymethylation, is an important epigenetic switch in the development of cervical cancer.” It is well known that the demethylation process is complex and involves multiple stages, including hydroxymethylation, aldehyde methylation, and carboxymethylation. Therefore, the authors should provide additional explanations about the viewpoint.

Reply: Indeed, as also mentioned and cited in the introduction, the demethylation process is a multistep process in which the enzymatic modification of 5-mC into 5-hmC is an initial step towards demethylation. We modified the text in the "Clinical utility" section of the discussion.

“When interpreting the result of such an analysis one has to realize, however, that the demethylation process not only involves a single step of hydroxymethylation but is a complex cascade that involves multiple stages, including aldehyde methylation and carboxymethylation, which could have a similar effect on this demethylation process. 

5. To evaluate differences in immunofluorescence staining results, it is recommended to use quantitative or semi-quantitative methods such as image analysis software instead of relying on visual inspection. Quantitative methods provide a more objective and reliable analysis of stain strength.

Reply: In our study we did not only rely on visual inspection alone. After tuning of the immunofluorescence protocols to obtain an optimal specificity and sensitivity, images were collected using a fixed integration time, allowing semiquantitative evaluation, and using the camera’s full dynamic range without signal intensity saturation. These images were then analyzed by visual inspection. In a previous study, we compared the result of visual inspection with quantitative microscopy of 5-hmC immunostaining patterns in normal cervical epithelium (Histochem Cell Biol…) and showed that differences between levels for 5-hmC within and between nuclei in tissue context could be analyzed. However, the complexity of the immunostaining patterns, particularly in CIN lesions, and the often scattered positive cells (as indicated above), only enabled a descriptive rather than a (semi)quantitative approach.

Reply to suggestions and comments of Reviewer #2:

1. Usually, during carcinogenesis, the methylation level around specific genes is increased” (page 3), it would be helpful to include the examples of aforementioned genes as well as to clarify whether they are associated to any particular types of cancer or are more general.

Reply: In the Introduction on page 3 we have included two new references [8,9} focusing on the role of gene methylation in several types of cancer.

“Numerous genes have been found to undergo hypermethylation in cancer. The genes that are susceptible are the genes involved in cell cycle regulation (e.g. p16INK4a, Rb), genes associated with DNA repair (e.g. BRCA1, apoptosis (DAPK, TMS1), drug resistance, detoxification, differentiation, angiogenesis, and metastasis. Although certain genes such as RASSF1A and p16 are commonly methylated in a variety of cancers, other genes are methylated in specific cancers [8.9]”. 

2. It was not specified how many healthy controls were examined, despite the experiments involving 65 cervical patients. 

Reply: No healthy controls were examined, normal histological areas present in the biopsies were analyzed. These areas were present in the patient samples and recognized by the pathologist and by molecular analysis. The normal areas showed to be HPV and p16 negative. We have added this text in the Materials and Methods under Table 1. 

3. Since the text and the table exhibited differences (65 vs. 60 respectively), the number of cervical samples needs to be clarified. (page 5)

Reply: 60 patients were analyzed, this was adapted in the text. 

4. The cervical grade abbreviations in Table 1 would be easier to understand if they were explained within the legend. (page 5)

Reply: We added CIN: cervical intraepithelial lesion, AIS: adenocarcinoma in situ; AdC: adenocarcinoma, SCJ: normal squamocolumnar junction (transition between squamous epithelium and glandular/columnar epithelium, no preneoplasia present in junction. 

5. In my opinion, it was not required to reconsider the effects of antigen retrieval protocols for the discovery of epigenetic DNA modifications. The authors have previously stated the benefit of citrate epitope unmasking in preserving cell nucleus morphology during immunofluorescence detection of 5-mC and 5-hmC (Moshi et al., Histochem Cell Biol 2023). (pages 5-6)

Reply: We have now omitted Supplemental Figure S1 and Supplemental Figure S4 (old figure numbers) and the antigen retrieval protocols for TE and pepsin/HCl in the Materials & Methods section because these have indeed been described earlier.

6. Ensure that the reagent names are accurate, please double-check them, e.g., buffer PBT or PBST (page 6)

Reply: We have corrected and checked the document for PBS and PBST.

7. As it was discussed above, authors have previously stated the benefit of citrate epitope unmasking in preserving cell nucleus morphology during immunofluorescence detection of 5-mC and 5-hmC (Moshi et al., Histochem Cell Biol 2023).

Therefore, I do not think it is essential to include this chapter and supplemental figure S1 (chapter: Choice for the citrate pH 6.0 retrieval protocol for the detection of 5-mC and 5-hmC in FFPE tissue sections. – page 8, Supplemental Figure S1 – page 29) as well as the text: “Also, when using TE and pepsin/HCl as antigen … was still clearly visible” and supplemental figure S4 (chapter: Immunostaining results of 5-mC and 5-hmC in premalignant endocervical glandular lesions – page 18, Supplemental Figure S4 – page 32) in the manuscript.

Reply: As stated above these Figures and the text in the Materials and Methods sections has been modified.

8. The headline of the chapter: “5-mC and 5-hmC immunostaining levels in normal ectocervical squamous epithelium and metaplastic (Figures 1-3)” is not accurate since authors included information about the metaplastic epithelium. The benign condition known as squamous metaplasia of uterine cervix presents no risk of malignancy. However, it cannot be defined as a normal state. (chapter: 5-mC and 5-hmC immunostaining levels in normal ectocervical squamous epithelium (Figures 1-3) – page 8)

Reply: We have changed the title of the chapter into 5-mC and 5-hmC immunostaining levels in normal squamous epithelium. As now indicated in the legend we defined metaplasia as normal epithelium since metaplasia is the transition of endocervical epithelium into squamous epithelium, a normal process that can be detected in cervical biopsies. We have modified the text as follows:

“We defined metaplasia as normal epithelium since it represents the transition of endocervical glandular epithelium into squamous epithelium, a normal physiological process that can be detected in cervical biopsies. Metaplastic epithelium characterizes the transformation zone and bridges the epithelium between the original and new SCJ. Fig 3A shows a stretch of (immature) metaplastic epithelium with the typical co-expression of SOX2 and SOX17 (Figs 3B and C, respectively), and at the same time a negative p16 staining (Fig 3D), indicating that the epithelium was not infected by HPV. For an overview showing the transition in these immunostaining patterns between normal and metaplastic epithelium we refer to S1 Fig.” 

9. The squamous epithelium is formed by multiple layers, such as the basal, parabasal, intermediate and superficial. To make the results more readable, I would consider marking the layer distribution within stained tissue. (Figure 1 – page 9-10)

Reply: We agree and modified the figures by indicating the different cellular structures to help reading the images. 

10. The results for 5-hmC expression in Figures 1D and 1G are confusing. Similar layer organization and, hence, corresponding 5-hmC labeling patterns should be found in the squamous epithelium represented by boxes 1 and 2. (Figure 1 – page 9-10)

Reply: As indicated in the text referring to Figure 1 the distance between the squamous epithelium in boxes 1 and 2 and the squamocolumnar junction apparently has an impact on the immunohistochemical distribution patterns, especially in the basal compartment. For that reason, we also studied normal squamous epithelium close to the SCJ (Figure 2), and two types of 5-hmC patterns can indeed be recognized. 

11. What was the rationale behind selecting and presenting samples from four separate patients. (Figure 4 – page 13)

Reply: Figure 4 has been reduced in size and now illustrates the most frequently occurring immunostaining patterns for 5-mC and 5-hmC in normal glandular epithelium.

12. Would suggest adding arrows to indicate where 5-mC and 5-hmC signals are seen/ not seen within reserve cells. (Figure G, H, J – page 14)

Reply: We modified the Figures and added RC and CC. 

13. The signatures of 5-mC/5-hmC/DAPI are missing within Figure 7A. (Figure 7A – page 17).

 Reply: We added the signatures of 5-mC/5-hmC/DAPI in the figure. Figure 7 was reorganized and labels were included for clarification.

14. The headlines in the results chapters might be altered to eliminate the phrase “figure” (page 8-18). 

Reply: We removed the phrase “figure” in all Chapter headings.

Reply to suggestions and comments of Reviewer #3:

The Authors just made a simple comparison of localization of the epigenetic changes in the selected samples, without any quantitative analysis. The pictures are nice, but they lack significance without showing results of the real data analysis (retrieved possibly from optical density values).

Reply: In our study we could not rely on quantitative analyses because of the complexity of the immunostaining patterns, particularly in CIN lesions, and the often scattered positive cells (as indicated above). This enabled only a descriptive rather than a (semi)quantitative approach. The text in the results section, in particular the part for the CIN lesions, has been amended to better indicate the heterogeneity of these tissues and the complexity of the immunostaining patterns. Also, we have included new data from bright field analyses of AIS showing a clear down regulation of 5-hmC in these lesions (see Fig 7).

Complex immunostaining patterns for 5-mC and 5-hmC in squamous intraepithelial lesions 

The patient diagnosis as shown in Table 1 was based on the highest histological CIN grade observed in the tissue sample, but virtually all samples contained areas with different grades of CIN. This was especially obvious in the higher-grade lesions. Using H&E staining of the tissue sections of the different patient samples as a guide to recognize specific areas with different grades of CIN, a correlation between the CIN stages with the different immunohistochemical (hydroxy)methylation patterns was made. From these analyses it became clear that with progression of the lesion the immunostaining patterns became more complex. Fig 6 summarizes these different immunostaining patterns recognized for 5-mC and 5-hmC in areas marked by the pathologist as CIN1, CIN2 and CIN3. In the CIN1 lesions we observed similar patterns for 5-mC and 5-hmC as recorded in the normal squamous epithelium (Fig 6A-C), with intense to weak staining in the basal/parabasal compartment and lesser staining in the intermediate and superficial layers. Both 5-mC and 5-hmC positive nuclei intermingled with weakly stained and negative nuclei. A pattern that was observed in nearly all higher grade CIN2-3 lesions, several of which also contained areas with a CIN1 morphology, is illustrated in Figs 6D-F, exhibiting weak to negative staining in the lower compartment, but more reactivity in the upper layers. Most CIN3 areas showed this pattern with a weak staining in the lower compartment of the epithelium, but areas diagnosed as CIN2/3 or CIN3 occasionally showed overlapping immunostaining for both 5-mC and 5-hmC in all epithelial layers (Figs 6 G and H). Finally, in about 20% of the patients diagnosed with CIN3 (5 out of 25 patients), a (very) weak staining for 5-hmC was observed throughout the entire epithelium (Fig 6K), while most nuclei were positive for 5-mC (Fig 6J). In summary, CIN lesions showed a heterogeneous staining pattern for 5-hmC. When studying CIN3 cases in more detail, combinations of the patterns described above were found next to each other (see S2 Fig for examples). Also, when performing confocal microscopy, CIN lesions were found to show areas where 5-mC and 5-hmC immunostainings were mutually exclusive, with nuclei exhibiting intense staining for 5-hmC, while being negative for 5-mC and vice versa (see S3 Fig).

---

## [Decision Letter · Decision Letter 1]

11 Sep 2023

PONE-D-23-13996R1Inhibition of cytosine 5-hydroxymethylation during progression of cancer precursor lesions in the uterine cervixPLOS ONE

Dear Dr. Hopman,

Thank you for submitting your manuscript to PLOS ONE. After careful consideration, we feel that it has merit but does not fully meet PLOS ONE’s publication criteria as it currently stands. Therefore, we invite you to submit a revised version of the manuscript that addresses the points raised during the review process.

We look forward to receiving your revised manuscript.

Kind regards,

Ricardo Ney Oliveira Cobucci, Ph.D

Academic Editor

PLOS ONE

**Additional Editor Comments:**

Dear authors, one of the reviewers still requests that you review the manuscript and we ask that you pay attention to the recommendations made.

Reviewers' comments:

Reviewer's Responses to Questions

**Comments to the Author**

1. If the authors have adequately addressed your comments raised in a previous round of review and you feel that this manuscript is now acceptable for publication, you may indicate that here to bypass the “Comments to the Author” section, enter your conflict of interest statement in the “Confidential to Editor” section, and submit your "Accept" recommendation.

Reviewer #1: All comments have been addressed

Reviewer #2: All comments have been addressed

2. Is the manuscript technically sound, and do the data support the conclusions?

Reviewer #1: Partly

Reviewer #2: Yes

3. Has the statistical analysis been performed appropriately and rigorously? 

Reviewer #1: N/A

Reviewer #2: N/A

4. Have the authors made all data underlying the findings in their manuscript fully available?

Reviewer #1: Yes

Reviewer #2: Yes

5. Is the manuscript presented in an intelligible fashion and written in standard English?

Reviewer #1: No

Reviewer #2: Yes

6. Review Comments to the Author

Reviewer #1: Jobran et al.'s response addressed some of the questions, but there are still certain questions that require further clarification.

1. In Figure 1, in the normal tissue section, whether in the 5-mC staining group or the 5-hmC staining group, we can observe two types of cells: a group of cells with strong staining and a group of cells with weak staining. In Figure 6 and 7, in the abnormal tissue section (CIN), whether in the 5-mC staining group or the 5-hmC staining group, we can also observe groups of cells with strong and weak staining. Therefore, we are curious about the relationship between "staining intensity" and normal or abnormal cells. Additionally, the authors introduce the concept of "staining pattern". They descripted that "Intense staining is observed in the basal/parabasal layers, which differs in intensity from the upper layers in the part of the epithelium shown in box 1 (Fig 1D). The epithelium represented in box 2 and box 3 showed more intense 5-hmC staining in the intermediate and superficial cell layers compared to the basal/parabasal layers (Figs 1G and J)" (Page 10). We think that the description of "staining pattern" carries a strong subjectivity from the authors themselves. Can the "staining pattern" be further divided into different types? Is the "staining pattern" consistent across all samples (60 samples)? To what extent can the "staining pattern" replace the results of “HE staining”? We understand that scientific research requires quantitative descriptions. Results with excessive subjectivity make it difficult to accurately classify different stages of diseases. The authors need to delve into these important questions in order to answer the relationship between the "staining intensity" and "staining pattern" of 5-mC and 5-hmC with the cell types of the disease and disease progression.

2. In the figure legend of Fig 6, it is mentioned that the four groups of images correspond to "CIN1 area", "CIN1 area," "CIN3 area", and "CIN3 area". However, in page 15 the authors refer to "CIN2" and "CIN2/3" when describing Figure 6. The figure legend of Fig 6 does not correspond with the text of the manuscript. Further clarification is required to resolve this discrepancy.

3. Page 15. The text mentions "Using H&E staining of the tissue sections of the different patient samples as a guide to recognize specific areas with". The authors should provide H&E staining pathology images and describe the percentage of atypical cells.

4. Page 15. In the text, Figure 6A-C is described as "In the CIN1 lesions, we observed similar patterns for 5-mC and 5-hmC as recorded in the normal squamous epithelium (Fig 6A-C), with intense to weak staining in the basal/parabasal compartment and lesser staining in the intermediate and superficial layers", in which "basal/parabasal compartment, intermediate and superficial layers" correspond to basal cells (bc), parabasal cells (pbc), intermediate cell layers (icl) in figure legend, respectively.

However, the author mentioned "lesser staining in the superficial layers" in the text, while the figure legend states "The superficial cell layer is not shown in the fluorescence images." Please provide additional images that include the "superficial cell layer." This will help ensure consistency between the text and the figure legend and provide a more complete representation of the study's findings.

5. Page 15. Please clarify how to obtain the result that "Most CIN3 areas showed this pattern with weak staining in the lower compartment of the epithelium" of Figs 6 G and H. And the authors should display the immunostaining results for all 5 patients diagnosed with CIN3. Please label the images clearly to indicate their source from each of the five patients. This will help readers understand and assess the consistency of the observed staining patterns across the CIN3 cases.

6. Page 15. The author mentioned, "while most nuclei were positive for 5-mC (Fig 6J)", but it is not clear from Figure 6J that the nuclei of the cells are positive. The authors should provide clearer and more representative images or data that demonstrate the positive staining of nuclei for 5-mC. This will help ensure that the findings are accurately represented and can be readily interpreted by readers.

7. Page 26. In the "Clinical utility" section of the Discussion, the authors modified the text as "When interpreting the result of such an analysis one has to realize, however, that the demethylation process not only involves a single step of hydroxymethylation but is a complex cascade that involves multiple stages, including aldehyde methylation and carboxymethylation, which could have a similar effect on this demethylation process [2]".

As mentioned in the reference you cited, it might be more accurate to state that "5-hydroxymethylcytosine (5hmC), 5-formylcytosine (5fC), and 5-carboxylcytosine (5caC)" compared to "cytosine hydroxymethylation", "aldehyde methylation" and "carboxymethylation".

Reviewer #2: My suggestions have been implemented. The manuscript discusses the potential prognostic significance of (hydroxy)methylation for progression of cancer precursor lesions in the uterine cervix. I believe the work is worthy of publishing.

7. PLOS authors have the option to publish the peer review history of their article (what does this mean?). If published, this will include your full peer review and any attached files.

Reviewer #1: No

Reviewer #2: No

---

## [Author Response · Author response to Decision Letter 1]

26 Oct 2023

Rebuttal to the Reviewer’s Comments 

First of all, we would like to thank the reviewer for the meticulous assessment of our data and our interpretation of the results. We have therefore made every effort to amend the parts of the manuscript that were commented on. Several new figures now replace the former ones, some of which are included in the Supplemental information paragraph. Also, the text has been adapted and extended to respond to the questions and suggestions of this reviewer.

We can reply to the reviewer’s comments as follows:

Question 1A: In Figure 1, in the normal tissue section, whether in the 5-mC staining group or the 5-hmC staining group, we can observe two types of cells: a group of cells with strong staining and a group of cells with weak staining. In Figure 6 and 7, in the abnormal tissue section (CIN), whether in the 5-mC staining group or the 5-hmC staining group, we can also observe groups of cells with strong and weak staining. Therefore, we are curious about the relationship between "staining intensity" and normal or abnormal cells. 

Answer: We have now performed quantitative analyses of the immunostaining in nuclei from normal and neoplastic cervical epithelium to compare the staining intensities of normal and abnormal cells. These data have been included in the Supplemental information paragraph as S3 Fig and the results have been described in the legend to the figure and summarized in the paragraph dealing with the squamous intraepithelial lesions. It should be realized that comparison of staining intensities amongst different tissues (normal versus neoplastic) are difficult because of the different fixation conditions of the tissues, the accessibility of the antigens, etc. That is why these quantitative analyses can only be used to compare the staining intensities between cells in the same tissue region. The quantitative data can, however, be used to support the semiquantitative observations that largely three different 5-hmC distribution patterns can be distinguished in normal and neoplastic cervical epithelium.

Question 1B: Additionally, the authors introduce the concept of "staining pattern". They descripted that "Intense staining is observed in the basal/parabasal layers, which differs in intensity from the upper layers in the part of the normal (?) epithelium shown in box 1 (Fig 1D). The epithelium represented in box 2 and box 3 showed more intense 5-hmC staining in the intermediate and superficial cell layers compared to the basal/parabasal layers (Figs 1G and J)" (Page 10). We think that the description of "staining pattern" carries a strong subjectivity from the authors themselves. Can the "staining pattern" be further divided into different types? Is the "staining pattern" consistent across all samples (60 samples)? To what extent can the "staining pattern" replace the results of “HE staining”? We understand that scientific research requires quantitative descriptions. Results with excessive subjectivity make it difficult to accurately classify different stages of diseases. The authors need to delve into these important questions in order to answer the relationship between the "staining intensity" and "staining pattern" of 5-mC and 5-hmC with the cell types of the disease and disease progression.

Answer: We have taken several steps to refute the image of subjectivity when it comes to the “staining pattern” terminology.

- Firstly, at the beginning of the Results section we have now included a short paragraph to explain the difference and relationship between the terms "staining intensity" and "staining pattern" of 5-mC and 5-hmC in normal and preneoplastic tissues. 

- Secondly, we refer to our earlier study (reference 25) that describes the staining and distribution patterns of 5-mC and 5-hmC in normal cervical squamous epithelium. This is a study in which we tested several retrieval protocols for an optimal detection of 5-mC and 5-hmC in paraffin sections and performed already quantitative analyses of the staining intensities of cells in the different layers of the normal cervical epithelium.

- Thirdly, in the Results paragraph dealing with the squamous intraepithelial lesions the 3 major types of 5-hmC staining patterns (now designated as distribution patterns) are specified. 

- Fourthly, these distribution patterns are not consistent across all 60 patient samples and therefore we have included the number of tissue areas that show the individual patterns. This also replies to the reviewer’s comment urging for a quantitative description to answer the relationship between the staining pattern of 5-mC and 5-hmC with the cell types of the disease and disease progression.

The H&E staining cannot be replaced by the 5-mC or 5-hmC staining pattern. In our study the H&E staining is used as the golden standard for the diagnosis of the CIN lesions in combination with more conventional immunomarkers such as p16, Ki-67, Cytokeratins, SOX2 and SOX17.

Question 2: In the figure legend of Fig 6, it is mentioned that the four groups of images correspond to "CIN1 area", "CIN1 area," "CIN3 area", and "CIN3 area". However, in page 15 the authors refer to "CIN2" and "CIN2/3" when describing Figure 6. The figure legend of Fig 6 does not correspond with the text of the manuscript. Further clarification is required to resolve this discrepancy.

Answer: We have now included a new Figure 6 into the body of the text and moved the former Figure 6 to the Supplemental information paragraph as S2 Fig. The new Figure 6 shows H&E and immunofluorescence staining images for 5-mC and 5-hmC in a CIN1, a CIN2 and two different patterns for CIN3. Also the body of the text and the legend to figure 6 has been adapted to clarify the apparent discrepance that existed in the former version of the manuscript.

Question 3: Page 15. The text mentions "Using H&E staining of the tissue sections of the different patient samples as a guide to recognize specific areas with". The authors should provide H&E staining pathology images and describe the percentage of atypical cells.

Answer: H&E staining images are now incorporated into the new Figure 6 for a CIN1, a CIN2 and two different patterns for CIN3. The number of atypical cells, as well as the dysplastic appearance increases with the CIN grade and was used as a diagnostic criterium by the pathologist.

Question 4: Page 15. In the text, Figure 6A-C is described as "In the CIN1 lesions, we observed similar patterns for 5-mC and 5-hmC as recorded in the normal squamous epithelium (Fig 6A-C), with intense to weak staining in the basal/parabasal compartment and lesser staining in the intermediate and superficial layers", in which "basal/parabasal compartment, intermediate and superficial layers" correspond to basal cells (bc), parabasal cells (pbc), intermediate cell layers (icl) in figure legend, respectively.

However, the author mentioned "lesser staining in the superficial layers" in the text, while the figure legend states "The superficial cell layer is not shown in the fluorescence images." Please provide additional images that include the "superficial cell layer." This will help ensure consistency between the text and the figure legend and provide a more complete representation of the study's findings.

Answer: We have now included a new Figure 6 that contains lower magnifications of immunofluorescence staining patterns for 5-mC and 5-hmC for the different grades of CIN. These images now also include the superficial cell layers. The former Figure 6 with the higher magnification images (and therefore missing the superficial cell layers) is now S2 Fig in the Supplemental information paragraph.

Also, in Figure 1 we have now included lower magnifications of immunofluorescence staining patterns for 5-mC and 5-hmC for the normal cervical epithelium.

Question 5: Page 15. Please clarify how to obtain the result that "Most CIN3 areas showed this pattern with weak staining in the lower compartment of the epithelium" of Figs 6 G and H. And the authors should display the immunostaining results for all 5 patients diagnosed with CIN3. Please label the images clearly to indicate their source from each of the five patients. This will help readers understand and assess the consistency of the observed staining patterns across the CIN3 cases.

Answer: We believe that there exists some confusion about the number of patients diagnosed with CIN3. As can be seen from Table 1, of the 60 patients included in our study a total of 25 patients was diagnosed with CIN3. In the different neoplastic regions within the tissue sections different 5-hmc immunostaining patterns were observed. The observed numbers of these different patterns in relation to the total number of regions examined have now been included throughout the text. Only 5 of the 25 CIN3 lesions showed the pattern where all layers of the epithelium were negative for 5-hmC pattern. We see no use in depicting specifically these 5 cases with similar staining patterns in a separate image. In Figure 6 and in supplemental figure S2 Fig we have depicted the major 5-mC and 5-hmC staining patterns observed in the different CIN lesions, including an example of these negative cases.

Also, the text in the paragraph dealing with the squamous intraepithelial lesions has been adapted and extended, the quantitative data have been included and the different patterns have been more clearly described.

Question 6: Page 15. The author mentioned, "while most nuclei were positive for 5-mC (Fig 6J)", but it is not clear from Figure 6J that the nuclei of the cells are positive. The authors should provide clearer and more representative images or data that demonstrate the positive staining of nuclei for 5-mC. This will help ensure that the findings are accurately represented and can be readily interpreted by readers.

Answer: We have now included a new figure 6 with representative images that demonstrate the positive staining of nuclei for 5-mC and its relation to the 5-hmC and DAPI staining patterns. The low magnification images provide a good overview of the total tissue area and should now be easier to interpret by the readers.

The former Figure 6 is now supplementary figure S2 Fig. To show the overlap with the 5-mC staining we have now also slightly increased the intensity of the DAPI image in what was the former Figure 6L and is now supplementary figure S2L Fig.

Question 7: Page 26. In the "Clinical utility" section of the Discussion, the authors modified the text as "When interpreting the result of such an analysis one has to realize, however, that the demethylation process not only involves a single step of hydroxymethylation but is a complex cascade that involves multiple stages, including aldehyde methylation and carboxymethylation, which could have a similar effect on this demethylation process [2]".

As mentioned in the reference you cited, it might be more accurate to state that "5-hydroxymethylcytosine (5hmC), 5-formylcytosine (5fC), and 5-carboxylcytosine (5caC)" compared to "cytosine hydroxymethylation", "aldehyde methylation" and "carboxymethylation".

Answer: We have now amended the text in the Clinical utility paragraph as follows: 

“When interpreting the result of such an analysis one has to realize, however, that in addition to 5-hmC, the Tet proteins can generate 5-formylcytosine (5-fC) and 5-carboxylcytosine (5-caC) from 5-mC. A potential alternative 5-mC demethylation mechanism has therefore been suggested by Ito et al [2], in which the Tet proteins oxidize 5-mC not only to 5-hmC, but also to its aldehyde form 5-fC and the carboxylic acid form 5-caC.“

---

## [Decision Letter · Decision Letter 2]

8 Nov 2023

PONE-D-23-13996R2Inhibition of cytosine 5-hydroxymethylation during progression of cancer precursor lesions in the uterine cervixPLOS ONE

Dear Dr. Hopman,

Thank you for submitting your manuscript to PLOS ONE. After careful consideration, we feel that it has merit but does not fully meet PLOS ONE’s publication criteria as it currently stands. Therefore, we invite you to submit a revised version of the manuscript that addresses the points raised during the review process.

One of the reviewers even requests corrections to the manuscript.

We look forward to receiving your revised manuscript.

Kind regards,

Ricardo Ney Oliveira Cobucci, Ph.D

Academic Editor

PLOS ONE

Journal Requirements:

Additional Editor Comments:

Dear authors,

one of the reviewers even requests corrections to the manuscript.

Reviewers' comments:

Reviewer's Responses to Questions

**Comments to the Author**

1. If the authors have adequately addressed your comments raised in a previous round of review and you feel that this manuscript is now acceptable for publication, you may indicate that here to bypass the “Comments to the Author” section, enter your conflict of interest statement in the “Confidential to Editor” section, and submit your "Accept" recommendation.

Reviewer #1: (No Response)

Reviewer #2: All comments have been addressed

2. Is the manuscript technically sound, and do the data support the conclusions?

Reviewer #1: Partly

Reviewer #2: Yes

3. Has the statistical analysis been performed appropriately and rigorously? 

Reviewer #1: N/A

Reviewer #2: N/A

4. Have the authors made all data underlying the findings in their manuscript fully available?

Reviewer #1: Yes

Reviewer #2: Yes

5. Is the manuscript presented in an intelligible fashion and written in standard English?

Reviewer #1: No

Reviewer #2: Yes

6. Review Comments to the Author

Reviewer #1: Jobran et al.'s responses addressed some of the questions, but there are still certain questions that require further clarification.

In the "Clinical utility" section, "As a consequence... the detection of global hydroxymethylation by means of immunohistochemical staining of 5-hmC may potentially become an important diagnostic tool in the investigation of glandular lesions of the cervix ", The authors concluded that immunohistochemical staining of 5-hmC can be used as an important diagnostic tool for cervical glandular lesions.

The 5-hmC staining technique used in this study can be divided into strong staining and weak staining in principle. In addition, the samples in this study were divided into 5 groups according to HE staining results, including normal squamous epithelium, normal columnar epithelium and reserve cells, squamous intraepithelial lesions, premalignant endocervical glandular lesions, cervical adenocarcinomas. The authors then described 5-hmC staining patterns (staining intensity and staining distribution) in these samples, including negative, weak, intermediate or intense/strong (see the attached figure). The authors also agree that the 5-hmC staining patterns cannot replace HE staining results. Therefore, the authors believe that immunohistochemical staining of 5-hmC can be used as a diagnostic tool for cervical glandular lesions, an important conclusion that more evidence is needed to support. For example: (1) the proportion of 5-hmC staining pattern in different disease samples (the attached figures showed that the proportion of 5-hmC staining strong was not high in some disease samples); (2) 5-hmC staining pattern sensitivity and specificity of different diseases using ROC/AUC value comparing with the golden standard HE method; (3) Individual differences in staining patterns judged by different experimenters. Therefore, it is recommended that authors need to add explanations to this issue or revise their conclusion to prevent misleading readers. Or add reasonable explanations in the discussion section to explain the shortcomings of this research work, so as to facilitate readers to understand the innovation points and potential application directions of this research work.

In addition, in "S3 fig R2, D-E-F", the bottom green signal is not measured, so there is no signal peak on the left in figure F. Please confirm whether the author missed this important signal?

Reviewer #2: All of my suggestions and comments have already been incorporated into the manusript.

Nonetheless, I believe the presented modifications substantialy improved the paper's quality.

7. PLOS authors have the option to publish the peer review history of their article (what does this mean?). If published, this will include your full peer review and any attached files.

Reviewer #1: No

Reviewer #2: No

---

## [Author Response · Author response to Decision Letter 2]

21 Dec 2023

Rebuttal to the Reviewer’s Comments 

We would like to thank the reviewer for the meticulous assessment of our revision. The pdf entitled: Figure_integration.pdf provides an excellent overview of the 5-hmC staining patterns in relation to the type of (premalignant) epithelia. We would appreciate that the peer review history would be included with our manuscript, including the Figure_integration.pdf from the reviewer. 

We can reply to the editor’s and reviewer’s comments as follows:

Question: Please review your reference list to ensure that it is complete and correct.

Answer: We have reviewed the reference list to ensure that it is complete and correct.

Question Reviewer #1: Jobran et al.'s responses addressed some of the questions, but there are still certain questions that require further clarification.

In the "Clinical utility" section, "As a consequence... the detection of global hydroxymethylation by means of immunohistochemical staining of 5-hmC may potentially become an important diagnostic tool in the investigation of glandular lesions of the cervix ", The authors concluded that immunohistochemical staining of 5-hmC can be used as an important diagnostic tool for cervical glandular lesions……. Therefore, it is recommended that authors need to add explanations to this issue or revise their conclusion to prevent misleading readers. Or add reasonable explanations in the discussion section to explain the shortcomings of this research work, so as to facilitate readers to understand the innovation points and potential application directions of this research work.

Answer: We have adapted the text in the “Clinical utility” section and revised our conclusion as follows. 

As a consequence, not only the methylation status of specific genes can be used as an indicator for the degree of malignant transformation of the cervical squamous and glandular lesions [15-17], but also the detection of global hydroxymethylation by means of immunohistochemical staining of 5-hmC. In case of glandular lesions an inhibition of global demethylation was frequently recognized in AIS, the pre-stage of adenocarcinoma, and may become of clinical utility in combination the conventional H&E staining. Such a correlation between the loss of 5-hmC levels and the degree of malignant progression was less evident for the squamous premalignant lesions. For that reason, the potential clinical utility of 5-mC and 5-hmC immunostaining is limited for these squamous lesions. In diagnostically challenging clinical CIN samples, however, the 5-hmC staining patterns, showing either positivity throughout the full thickness of the epithelium or a negative epithelium, are indications for a high-grade lesion in addition to the morphological criteria. 

Question Reviewer #1: In addition, in "S3 fig R2, D-E-F", the bottom green signal is not measured, so there is no signal peak on the left in figure F. Please confirm whether the author missed this important signal?

Answer: We included a note in the legend of Figure S3: Note that the stromal compartment always shows cells with a high nuclear intensity staining for 5-hmC (see also Figs 1 and 2), which are localized underlying the epithelial basal cell compartment (see S3 Fig D). These stromal cells have not been included in this quantitative analysis (see S3 Figs E and F). 

For the reader we also have added this note to the legends of Figs 1 and 2: Note also the intense 5-hmC immunostaining in the stromal cell compartment.

---

## [Decision Letter · Decision Letter 3]

28 Dec 2023

Inhibition of cytosine 5-hydroxymethylation during progression of cancer precursor lesions in the uterine cervix

PONE-D-23-13996R3

Dear Dr. Hopman,

We’re pleased to inform you that your manuscript has been judged scientifically suitable for publication and will be formally accepted for publication once it meets all outstanding technical requirements.

Kind regards,

Ricardo Ney Oliveira Cobucci, Ph.D

Academic Editor

PLOS ONE

Additional Editor Comments (optional):

Congratulations.

Reviewers' comments:

Reviewer's Responses to Questions

**Comments to the Author**

1. If the authors have adequately addressed your comments raised in a previous round of review and you feel that this manuscript is now acceptable for publication, you may indicate that here to bypass the “Comments to the Author” section, enter your conflict of interest statement in the “Confidential to Editor” section, and submit your "Accept" recommendation.

Reviewer #1: All comments have been addressed

2. Is the manuscript technically sound, and do the data support the conclusions?

Reviewer #1: Yes

3. Has the statistical analysis been performed appropriately and rigorously? 

Reviewer #1: Yes

4. Have the authors made all data underlying the findings in their manuscript fully available?

Reviewer #1: Yes

5. Is the manuscript presented in an intelligible fashion and written in standard English?

Reviewer #1: No

6. Review Comments to the Author

Reviewer #1: (No Response)

7. PLOS authors have the option to publish the peer review history of their article (what does this mean?). If published, this will include your full peer review and any attached files.

Reviewer #1: No

---

## [Editor Report · Acceptance letter]

25 Jan 2024

PONE-D-23-13996R3 

PLOS ONE

Dear Dr. Hopman, 

I'm pleased to inform you that your manuscript has been deemed suitable for publication in PLOS ONE. Congratulations! Your manuscript is now being handed over to our production team.

Kind regards, 

on behalf of

PROFESSOR Ricardo Ney Oliveira Cobucci 

Academic Editor

PLOS ONE